# A Case Study for the Recovery of Authentic Microbial Ancient DNA from Soil Samples

**DOI:** 10.3390/microorganisms10081623

**Published:** 2022-08-10

**Authors:** Vilma Pérez, Yichen Liu, Martha B. Hengst, Laura S. Weyrich

**Affiliations:** 1Australian Centre for Ancient DNA (ACAD), School of Biological Sciences, University of Adelaide, Adelaide, SA 5005, Australia; 2ARC Centre of Excellence for Australian Biodiversity and Heritage (CABAH), School of Biological Sciences, University of Adelaide, Adelaide, SA 5005, Australia; 3Key Laboratory of Vertebrate Evolution and Human Origins, Institute of Vertebrate Paleontology and Paleoanthropology, Center for Excellence in Life and Paleoenvironment, Chinese Academy of Sciences, Beijing 100044, China; 4Laboratorio de Ecología Molecular y Microbiología Aplicada, Departamento de Ciencias Farmacéuticas, Facultad de Ciencias, Universidad Católica del Norte, Antofagasta 1270300, Chile; 5Department of Anthropology and Huck Institutes of the Life Sciences, The Pennsylvania State University, State College, PA 16802, USA

**Keywords:** environmental genomics, ancient DNA, sedaDNA, soil, paleomicrobiome

## Abstract

High Throughput DNA Sequencing (HTS) revolutionized the field of paleomicrobiology, leading to an explosive growth of microbial ancient DNA (aDNA) studies, especially from environmental samples. However, aDNA studies that examine environmental microbes routinely fail to authenticate aDNA, examine laboratory and environmental contamination, and control for biases introduced during sample processing. Here, we surveyed the available literature for environmental aDNA projects—from sample collection to data analysis—and assessed previous methodologies and approaches used in the published microbial aDNA studies. We then integrated these concepts into a case study, using shotgun metagenomics to examine methodological, technical, and analytical biases during an environmental aDNA study of soil microbes. Specifically, we compared the impact of five DNA extraction methods and eight bioinformatic pipelines on the recovery of microbial aDNA information in soil cores from extreme environments. Our results show that silica-based methods optimized for aDNA research recovered significantly more damaged and shorter reads (<100 bp) than a commercial kit or a phenol–chloroform method. Additionally, we described a stringent pipeline for data preprocessing, efficiently decreasing the representation of low-complexity and duplicated reads in our datasets and downstream analyses, reducing analytical biases in taxonomic classification.

## 1. Introduction

Paleomicrobiological studies of ancient microbial molecules (such as DNA, proteins, or lipids) provide insights into how microbial species, populations, and ecosystems evolve over time [1,2,3]. Nowadays, paleomicrobiologists increasingly rely on the recovery of ancient DNA (aDNA)—a highly degraded, fragmented, and chemically modified DNA (e.g., deamination at the 5′ and 3′ ends) extracted from historical, archaeological, and palaeoecological remains [4]. Reconstructed ancient microbiomes can also be linked and compared to living microbial populations in similar, modern ecologies to study past biological and ecological shifts.

Ancient DNA can be readily recovered from deposited sediments, thereby providing novel insights into past microbial communities of diverse environmental niches [5,6,7,8], and represents unexplored repositories of the past microbial life and the environmental conditions present at the time. Several studies successfully recovered aDNA from non-sediment sample types. For example, Frisia and colleagues [9] recovered potential thermophilic microbial aDNA from subglacial calcite crusts; alongside petrographic and geochemical records, providing evidence to support past episodes of Antarctic volcanism that influenced ocean productivity. Moreover, Turney et al. [10] recovered microbial aDNA and marine biomarkers (using Liquid Chromatography-Organic Carbon Detection, LC-OCD; Imaging Flow Cytometry, IFC analysis, and fluorescent organic matter, fOM) from Antarctic ice cores to explore the environment during the Antarctic Cold Reversal (14.6–12.7 kyr BP). This study suggested that enhanced marine biological productivity and increased CO_2_ sequestration occurred at this time. Further, Thomas and colleagues [11] investigated past climate and environmental change in the eastern Australian Highlands using a multiproxy framework including pollen and charcoal analysis, high-resolution geochemistry, and ancient microbial community composition from sediment samples of Club Lake. The findings suggested that the general warming trend over the late Holocene in this region was concurrent with fire activity and ecosystem shifts. The previously presented studies show how microbial aDNA can provide information on past geological events, climate change, and shifts in biodiversity.

Despite the early use of aDNA in environmental studies, most aDNA studies have not yet optimized their DNA extraction protocols for microbial aDNA, verified the accurate representation of complex, ancient soil microbial communities, or examined background contaminant DNA and cross-contamination that can plague ancient microbial studies [12,13]. Therefore, there are currently considerable gaps in our knowledge regarding the most appropriate laboratory protocols and bioinformatics pipelines in the environmental paleomicrobiology field, although a few environmental [8,14,15] and non-environmental paleomicrobiological studies exploring these gaps do exist [16,17,18,19]. The characteristic damage and extensive fragmentation of aDNA influence not only the selection of molecular techniques [20] but also bioinformatic tools used to analyze the sequencing data [17,18,19,21], which is more complex due to the short sequencing reads [22]. Therefore, it is essential to examine and benchmark resources of laboratory and bioinformatic components employed in the analyses of ancient environmental microbiomes to help reduce noise within the dataset and validate the results.

Here, we explored these gaps by reviewing up-to-date guidelines for human, animal, and plant aDNA studies and integrated these concepts into a case study on ancient soil microbiomes. We compared five different extraction methods and employed a shotgun metagenomic sequencing approach to examine environmental DNA in soil samples of two extreme environments of the Chilean Andes. We considered both ecosystems as model systems for paleogenomic studies for several reasons, namely (1) the availability of crucial information from present and ancient microbial communities inhabiting extreme environments; (2) the physicochemical conditions; and (3) high rates of mineralization, which may favor microbial DNA preservation [23]. We also compared bioinformatic pipelines to process ancient shotgun sequencing data, including data preprocessing, taxonomic classification, and aDNA authentication, and tested eight bioinformatic pipelines to determine the current best practices to recover and analyze environmental microbial aDNA. We examined the impact of different parameters in these processing pipelines using the twenty nine datasets (twenty datasets for the samples and nine for extraction blank controls—EBCs) generated with different extraction methods and examined how fragment size, number of collapsed reads, taxonomic composition (i.e., archaea, bacteria, and eukaryotes), and damage profiles vary between the datasets. Overall, the discussion of the methods presented in this case study aimed to provide a resource for scientists who are starting ancient microbiome analyses and generate discussion about the accurate, reproducible recovery of authentic microbial aDNA from environmental samples.

## 2. Materials and Methods

### 2.1. Sample Location

Terrestrial soil samples were collected from the following two extreme environments in the Chilean Altiplano: the Lirima hydrothermal system and the Huasco wetland (Figure 1). The Lirima hydrothermal system (19°51′24″ S, 68°55′02″ W) is located 25 km SW of the Sillaguay volcanic chain near the Aroma-Quimchasata volcanic complex (Tarapacá region, Chile) over 4000 m a.s.l. [24,25,26]. Huasco wetland (Tarapacá region, Chile) (20°17′7″ S, 68°50′7″ W) is a high-altitude wetland (3800 m a.s.l.) with extreme environmental conditions (e.g., negative water balance, wide salinity range, and high daily temperature changes) [27,28]. Both sites are exposed to one of the highest solar radiation levels registered worldwide (>1200 W·m^−2^) [29,30], providing an unprecedented opportunity to examine how extremophilic microbes evolved over time.

### 2.2. Sample Collection and Storage Conditions

The sample collection, transportation, and storage were performed as suggested by Llamas et al. [31]. Briefly, during sample collection in the field, the researcher used a facemask, bodysuit, plastic boots, and gloves that were changed between each sample. All sampling equipment was pre-treated with 5% sodium hypochlorite and then ethanol, and a newly treated set of equipment was used for each sample collection.

A hole was excavated in each point to collect the samples. The outer layer of soil was removed (~1 cm) using a sterile spatula. The target soil was collected in a 50 mL sterile tube every 20 cm in a vertical transect from the bottom to the top of the excavation hole (Figure 1). For this study, we collected three different samples from the Lirima hydrothermal system, including S1 at 60 cm depth (19°51′4.3″ S 68°54′23.7″ W); S2 at 50 cm depth (19°51′4.2″ S 68°54′25.1″ W); and S3 at 110 cm depth (19°51′6.5″ S 68°54′24.2″ W), and a single sample from the Salar de Huasco system, S4 at 80 cm depth (20°17′13.6″ S 68°53′18.0″ W). Dating of soil samples was estimated using sedimentation rates of paleolakes in the central Altiplano of Bolivia, which are 70 km from both sites [32]. The sedimentation rate corresponds to approximately 1 mm yr^−1^, suggesting that our samples were between 500 and 1100 years old (S1:600 yo; S2:500 yo; S3: 1100 yo and S4:800 yo). 

Samples were immediately frozen using liquid nitrogen in the field. The tubes were individually wrapped in a sterile plastic bag and transported overnight with ice packs to the Australian Centre for Ancient DNA, University of Adelaide, Australia. Samples were frozen at −20 °C for six months before DNA extraction, as freezing may improve aDNA recovery from soils/sediments compared to fresh samples [33].

### 2.3. Sample Preparation in aDNA Facilities

We removed the outer 1 cm of soil surface and homogenized the internal soil by mixing. A subsample of 250 mg of homogenized soil was taken for DNA extraction. In addition, extraction blank controls (EBCs) were created by exposing tubes to air for 15 s before soil cores were decontaminated and were included at a rate of one EBC per every four soil samples. Specifically, we placed EBCs as the first (EBC1) and last tube (EBC2) of each extraction to help differentiate between laboratory background contamination (EBC1) and the impact of cross-contamination (EBC2) (Figure 1; E1: EBC1, E2: EBC2). 

### 2.4. DNA Extraction Methods in aDNA Facilities

In this study, we performed five different DNA extraction protocols to compare and select the best method for soil samples: (a)SiO_2_ + PowerLyzer kit (CP): performed using an in-house DNeasy^®^ PowerLyzer^®^ PowerSoil^®^ kit (Qiagen, Hilden, Germany) and silica-based method. We followed PowerLyzer kit protocol using 250 mg of soil samples up to step 10 of the manufacturer’s instructions (solution C3 and centrifugation). We then obtained the supernatant and used an in-house QG buffer (guanidine thiocyanate DNA-binding buffer) and a silica-based method, as previously described [17] with minor modifications. Briefly, we added the supernatant to 3 mL of binding buffer (2.8 mL QG buffer (Qiagen), 46 μL water, 15 μL NaCl (5M), 39 μL Triton-X 100 (Sigma-Aldrich, Saint Louis, MO, USA) and 167 μL acetic acid (3 M)) into a 15 mL tube, added 100 μL of medium-sized silica suspension and mixed under slow and constant rotation. We centrifuged the samples at 4500 rpm for 5 min and the pellet was washed twice by resuspension in 900 μL of 80% ethanol in a 1.5 mL tube. Tubes were centrifuged for 1 min at 14,000 rpm, and the supernatant was removed. The pellet was left to dry at 37 °C for 15 min, subsequently resuspended in 75 μL of pre-warmed (to 50 °C) TLE buffer (10 mM Tris-HCl, 1 mM EDTA), and incubated for 10 min. After pelleting for 1 min at 13,000 rpm, the supernatant was collected, aliquoted, and stored at −20 °C until further use.(b)SiO_2_ + PB buffer (PB): performed using a modified PB buffer (guanidine hydrochloride DNA-binding buffer) and silica-based method [34]. Briefly, 250 mg of soil samples was incubated overnight under slow, constant rotation at 37 °C in 1 mL of lysis buffer (900 μL EDTA; 80 μL SDS; 20 μL 20 mg/mL proteinase K). After lysis, samples were centrifuged at 14,500 rpm for 3 min, and the supernatant was added to 12.6 mL of binding buffer (12.2 mL PB buffer (Qiagen), 7 μL Tween-20, and 378 μL acetic acid (3 M)) in a 15 mL tube, with a 100 μL of medium-sized silica suspension and mixed under slow and constant rotation. We centrifuged the samples at 4500 rpm for 5 min, and the pellet was washed twice by resuspension in 900 μL of 80% ethanol in a 1.5 mL tube. Tubes were centrifuged for 1 min at 14,000 rpm, and the supernatant was removed. The pellet was left to dry at 37 °C for 15 min, subsequently resuspended in 75 μL of pre-warmed (to 50 °C) TLE buffer (10 mM Tris-HCl, 1 mM EDTA), and incubated for 10 min. After pelleting for 1 min at 13,000 rpm, the supernatant was collected, aliquoted, and stored at −20 °C until further use.(c)Phenol–chloroform (PHCH): completed as previously described [35] with minor modifications. Briefly, an equal volume of molecular grade phenol:chloroform was added to 250 mg of soil samples and mixed until an emulsion was formed. The solution was centrifuged at 12,000 rpm for 1 min at room temperature until both phases were separated. The aqueous phase was transferred to a fresh tube, and the process was repeated a second time. Then, an equal volume of chloroform was added, and the solution was mixed and centrifuged at 12,000 rpm for 1 min. The aqueous phase was transferred to a fresh tube. The DNA was precipitated by adding sodium acetate to a final concentration of 0.3 M and mixing the solution. Then, two volumes of ice-cold ethanol and 0.01 M of MgCl_2_ were added and mixed again. The solution was incubated at −20 °C for 30 min and centrifuged at 14,000 rpm for 15 min at room temperature. The supernatant was removed, and 1 mL of 70% ethanol was added. The solution was centrifuged at 14,000 rpm at 4 °C for 2 min, and the supernatant was removed. The tube was stored open in the hood at room temperature until all the ethanol was evaporated. The DNA pellet was redissolved in 75 μL of TLE buffer.(d)PowerLyzer kit (PL): DNA extraction of 250 mg of soil sample was performed using the DNeasy^®^ PowerLyzer^®^PowerSoil^®^ kit (Qiagen) and a Precellys 24 homogenizer (Bertin Instruments, Montigny-le-Bretonneux, France) according to the manufacturer’s instructions.(e)SiO_2_ + QG buffer (QG): performed using an in-house QG buffer (guanidine thiocyanate DNA-binding buffer) and silica-based method [17] with minor modifications. Briefly, 250 mg of soil sample were incubated overnight under slow, constant rotation at 37 °C in 1.72 mL lysis buffer (1.6 mL EDTA; 200 uL SDS; 20 uL 20 mg/mL proteinase K). After lysis, samples were centrifuged at 14,500 rpm for 2 min, and the supernatant was added to 3 mL of binding buffer (3.7 mL QG buffer (Qiagen), 61.4 μL water, 20 μL NaCl (5 M) and 52.1 μL Triton-X 100 (Sigma-Aldrich, Saint Louis, MO, USA) in a 15 mL tube, alongside 100 μL of medium-sized silica suspension. The solution was mixed under slow and constant rotation for one hour. We centrifuged the samples at 4500 rpm for 5 min and the pellet was washed two times by resuspension in 900 μL of 80% ethanol in a 1.5 mL tube. Tubes were centrifuged for 1 min at 14,000 rpm and the supernatant was removed. The pellet was left to dry at 37 °C for 15 min, resuspended in 75 μL of pre-warmed (to 50 °C) TLE buffer (10 mM Tris-HCl, 1 mM EDTA), and incubated for 10 min. After pelleting for 1 min at 13,000 rpm, the supernatant was collected, aliquoted, and stored at −20 °C until further use.

### 2.5. DNA Library Preparation and Shotgun Sequencing

After the DNA was extracted, double stranded shotgun metagenomic libraries were constructed for each extract, as previously described [36] with minor modifications. Briefly, 20 μL aliquot of DNA was repaired (15 min, 25 °C) using T4 DNA polymerase (New England Biolabs, Ipswich, MA, USA) in a 40 μL reaction. After purifying the repaired DNA using MinEluteTMReaction Cleanup Kit (Qiagen), truncated Illumina-adapter sequences containing two unique 5 base-pair (bp) barcodes were attached to the double-stranded DNA (60 min, 22 °C) using a T4 DNA ligase (Fermentas, Waltham, MA, USA). An additional DNA purification (MinEluteTM Reaction Cleanup Kit, Qiagen) step followed by a fill-in reaction with adapter sequences ((Bst DNA polymerase, New England Biolabs, Ipswich, MA, USA); 30 min, 37 °C, with polymerase deactivation for 10 min, 80 °C). Then, 5 μL of the reaction-product was used for a 25 μL PCR (three replicates per extract) with the primers IS7 and IS8. Each PCR reaction included 14.2 nuclease-free H_2_O, 2.5 μL 10× Gold Buffer, 2.5 μL 25 mM MgCl_2_, 0.25 μL 25 mM dNTPs, 1.25 IS7, 1.25 IS8, and 0.25 μL PlatinumTM Taq DNA polymerase High fidelity (ThermoFisher, Waltham, MA, USA). Thermal cycling specifications were as follows: 6 min at 94 °C; 13 cycles of 30 s denaturation at 94 °C, 30 s annealing at 60 °C, 40 s extension at 72 °C; and 10 min of final extension. We purified the PCR products using AxyPrep magnetic beads (Axygen Biosciences, Tewksbury, MA, USA; 1:1.8 library:beads) and eluted the DNA in Buffer EB (Qiagen) with 0.05% Tween^®^20 (Sigma Aldrich, Saint Louis, MO, USA) to remove primer-dimer. Then, a second PCR (eight replicated reactions per extract) was run using 2 μL of the purified DNA as template in a 25 uL reaction, following the same protocol as the previous PCR except for the use of Indexing primers IS4 y GAII index 1 [36] and 13 cycles. PCR products were purified using AxyPrep magnetic beads (Axygen Biosciences; 1:1.1 library:beads) and the DNA was eluted in Buffer EB (Qiagen) with 0.05% Tween^®^20 (Sigma Aldrich) to remove primer-dimers. The DNA quality and concentration was assessed using TapeStation (Agilent Technologies, Santa Clara, CA, USA). Each sequencing library was pooled together at equimolar concentrations. A series of 3 AxyPrep clean-ups (at a ratio of 1:1.1 of library:beads) and TapeStation runs were repeated until a sequencing pool was obtained with a minimal concentration of primer-dimer and sufficient DNA concentration (>1.5 nM) of the target library size range (i.e., 50–500 bp) for sequencing. The final pool was quantified using real-time quantitative PCR (qPCR) on an QuantStudio™ 6 Flex system (Applied Biosystems, Waltham, MA, USA) and submitted to the Garvan Institute of Medical Research, Sydney, Australia for Illumina HiSeqX (2 × 150 bp cycle). The targeted sequencing depth was 10 million reads per sample. 

### 2.6. Data Preprocessing

To compare the impact of bioinformatic analysis on the recovery of microbial aDNA, we tested two different pre-processing tools (AdapterRemoval2 [37] and Fastp [38]) to determine their ability to correctly quality-filter and collapse overlapping sequences. Further, we used different low-complexity read filter tools (FastP and Komplexity [39]), which examined three different low-complexity filter thresholds (30%, 55%, and 70%) and two sequence deduplication values (1: removal of exact sequences and 2: removal of exact sequences plus sequences with more than two mismatches). Overall, the raw data in the FASTQ format of 29 datasets (20 soil samples and 9 EBCs, as we could not recover DNA sequences from EBC2 extracted with phenol–chloroform) were pre-processed using 8 different pipelines, as follows: (1)Pre-filtered pipeline: the raw data were demultiplexed, adapter trimmed and merged using AdapterRemoval v.2.2.1 based on unique P5/P7 barcodes.(2)Post-filtered 30 kx: the raw data were demultiplexed, adapter trimmed and merged, using AdapterRemoval v.2.2.1. A low-complexity threshold of 30% was then applied using Komplexity v.0.3.6 followed by read deduplication that removed only exact sequences, using the dedupe tool of BBMap v.36.62 (https://sourceforge.net/projects/bbmap/; accessed on 1 December 2020).(3)Post-filtered 55 kx: the raw data were demultiplexed, adapter trimmed and merge using AdapterRemoval v.2.2.1. A low-complexity threshold of 55% was applied using Komplexity v.0.3.6 followed by read deduplication by removing exact sequences, using the dedupe tool of BBMap v.36.62.(4)Post-filtered 55 kx_2mm: the raw data were demultiplexed, adapter trimmed and merged using AdapterRemoval v.2.2.1. A low-complexity threshold of 55% was applied using Komplexity v.0.3.6 followed by reads deduplication by removing exact sequences and sequences with two mismatches, using the dedupe tool of BBMap v.36.62.(5)Post-filtered 70 kx: the raw data were demultiplexed, adapter trimmed and merged using AdapterRemoval v.2.2.1. A low-complexity threshold of 70% was applied using Komplexity v.0.3.6 followed by read deduplication by removing exact sequences, using the dedupe tool of BBMap v.36.62.(6)Post-filtered fastp_30 kx: the raw data were demultiplexed, adapter trimmed, merged, and a low-complexity threshold of 30% was applied, using Fastp v.0.20.0. Then, collapsed reads were deduplicated by removing exact sequences using the dedup tool of BBMap v.36.62.(7)Post-filtered fastp_30 kx_2mm: the raw data were demultiplexed, adapter trimmed, and merged, and a low-complexity threshold of 30% was applied, using Fastp v.0.20.0. Then, collapsed reads were deduplicated by removing exact sequences and sequences with two mismatches, using the dedupe tool of BBMap v.36.62.(8)Post-filtered fastp_55 kx: the raw data were demultiplexed, adapter trimmed, and merged, and a low-complexity threshold of 55% was applied, using Fastp v.0.20.0. Then, collapsed reads were deduplicated by removing exact sequences using the dedup tool of BBMap v.36.62.

We ran a FastQC (v.0.11.7; Babraham Bioinformatics [40]) quality control analysis on all pre-processed datasets (232 datasets: 160 from samples and 72 from the EBCs). The reports were visualized using MultiQC (v.1.0.dev0 [41]) and included read duplication %, GC%, read length, and the total number of sequences. To compare the performance of both types of pre-processing software, low-complexity filter thresholds, and deduplication values, we examined the number of collapsed reads, sequence length, and duplication levels in each case obtained from MultiQC. Statistical differences were compared by employing ANOVA (α < 0.05), followed by the Tukey post hoc test using R statistical software (R version 3.6.3).

### 2.7. Taxonomic Classification

We examined the impact of the five DNA extraction methods and eight bioinformatic pipelines on the taxonomic classification of DNA reads recovered from the soil samples. Since we used an alignment-based taxonomy classification, we also explored the impact of database choice on the reconstruction of soil microbiota, as it has been shown to bias the results significantly [19,42].

#### 2.7.1. Impact of DNA Extraction Methods and Bioinformatic Pipelines on Taxonomic Composition

The taxonomic composition of collapsed and non-collapsed reads from the 232 samples was determined using the MEGAN Alignment Tool (MALTn) v.0.3.8 [43]. DNA reads from datasets were aligned (default settings and semi-global alignment) against the SILVA SSU 132 Ref Nr99 database [44]. The resulting blast-text files were converted into RMA files using the blast2rma script included in MEGAN v.6.19.2 [45], using the following Last Common Ancestor (LCA) algorithm parameters: weighted-LCA (80%), minimum support of 0.05, minimum bit score of 50, minimum E-value of 0.01, and a minimum percent identity of 90%. RMA files were imported into MEGAN6 Community Edition (v.6.19.2 [45]) using the compare function as absolute read counts and ignoring unassigned reads to visualize taxonomic classifications. 

For analysis in QIIME2 [46], the reads from the 232 datasets were exported at the species-level into a BIOM format and imported into QIIME2 (v.2019.20). Decontam (v.1.10.0) analyses were carried out in R (v.3.6.3) on the BIOM file to help examine exogenous contamination in the EBCs [47]. We adopted a conservative approach and removed all species found in EBCs1 from the sample datasets to account for laboratory contaminants. Cross-contamination was explored in EBCs2, and species were removed from sample datasets if they were identified as contaminants (prevalence of 0.5 in datasets) in the Decontam analysis, using the function feature-table filter-features in QIIME2. Then, singletons were removed from the datasets using the same function. Communities were rarefied to 1000 species-identified sequences, and alpha (observed features and Shannon’s diversity indices) and beta (Jaccard and Bray–Curtis) diversity indices were calculated in datasets using the diversity core-metrics function of QIIME2 (Appendix A). Alpha and beta diversity differences among groups were tested using a Kruskal–Wallis analysis of variance and PERMANOVA, respectively, using the group_significance function in QIIME2 (v.2019.20) [46] and classing statistical insignificance for *p*-values > 0.05. Further, we used ANCOM differential abundance testing to identify microbial taxa driving compositional changes between contaminated and decontaminated datasets and between collapsed and non-collapsed reads in decontaminated datasets [48]. Beta diversity Principal Coordinate Analysis (PCoA) was plotted and visualized using the package ggplot2 in R (v.3.6.3) (Appendix A). Unassigned sequences were not considered in the statistical analysis. Taxonomic profiles of decontaminated datasets were plotted at the phylum level according to sample and Domain (Archaea, Bacteria, and Eukaryota) using the package ggplot2 in R (v.3.6.3).

#### 2.7.2. Impact of Databases Selection on Taxonomic Composition

After the final preprocessing pipeline was selected (Post-filtered 55 kx), we tested the following four different databases: SILVA SSU 132; archaeal and bacterial genomes at complete chromosome and scaffold-level from the RefSeq database June 2018, containing 47,713 archaeal and bacterial genome assemblies from the NCBI Assembly database [19]; NCBI nucleotide BLAST database downloaded on November 2019; and Genome Taxonomy Database (release 95) [49] to assess the impact of databases on the reconstruction of ancient bacterial communities in environmental samples, as reads assigned to the Bacteria domain represented more than 50% of the reads across all datasets. 

The taxonomic composition of collapsed reads of the 29 datasets preprocessed with the post-filtered_55kx pipeline was determined using the MEGAN Alignment Tool (MALTn) v.0.3.8 [43]. DNA reads were aligned (default settings and semi-global alignment) against each database. The resulting blast-text files were converted into RMA files using the blast2rma script included in the program MEGAN v.6.19.2 [45], following the lowest common ancestor (LCA) parameters: weighted-LCA (80%), minimum support of 0.05, minimum bit score of 50, minimum E-value of 0.01, and a minimum percent identity of 90%. To visualize taxonomic classifications, RMA files were imported into MEGAN6 Community Edition [45] using the compare function for absolute read counts and ignoring unassigned reads. 

For analysis in QIIME2 [46], the reads from the 29 datasets classified with each database were exported at species-level into a BIOM format and imported into QIIME2 (v.2019.20). A Decontam (v.1.10.0) analysis was carried out in R (v.3.6.3) on the BIOM file to help examine exogenous contamination in the EBCs. We adopted a conservative approach and removed all species found in EBCs, to account for laboratory contaminants. Cross-contamination was explored in EBCs2, and species were removed from sample datasets if they were identified as contaminants (prevalence of 0.5 in datasets in the Decontam analysis), using the function feature-table filter-features in QIIME2. Then, singletons were removed from the datasets using the same function. Only bacterial sequence reads from decontaminated datasets were kept for downstream analysis using the taxa filter-table function in QIIME2. For the taxonomic diversity analysis, datasets classified with the SILVA database were removed, as they contained fewer classified sequence reads than the rest of the databases (Appendix A). Communities were rarefied to 10,000 species-identified sequences to retain more datasets in the analysis (Appendix A). Alpha (observed features and Shannon’s diversity indices) and beta (Jaccard and Bray–Curtis) diversity indices were calculated in datasets using the diversity core-metrics function of QIIME2 (Appendix A). Alpha and beta diversity differences among groups were tested with a Kruskal–Wallis analysis of variance and PERMANOVA, respectively, using the group_significance function in QIIME2 (v.2019.20) [46] and statistical insignificance of *p*-values > 0.05. Beta diversity PCoA was plotted and visualized using the package ggplot2 in R (v.3.6.3) (Appendix A). Unassigned sequences were not considered in the statistical analysis. Taxonomic profiles of decontaminated datasets were plotted at the genus level using the package ggplot2 in R (v.3.6.3). 

### 2.8. Authentication of Microbial aDNA

To evaluate the authenticity of the results, we tested for DNA damage patterns using two different statistical models—Heuristic Operation for Pathogen Screening (HOPS) [50] and Changepoint [51]. For the screening of “ancient” taxa (i.e., taxa that presented damaged reads) with the HOPS software, we used collapsed reads from the 29 datasets preprocessed with the selected pipeline (Post-filtered 55 kx) and mapped against three different databases (RefSeq, NT, and GTDB). MaltExtract and post-processing functions (default parameters and a minimum percent identity of 90%) were run using a list of all taxa identified in the datasets at the species-level as target species for each reference database (129 species, Refseq; 5904 species, NT; 9716 species, GTDB) to maximize the screening. To simplify the discussion in this article, we selected the results obtained using GTDB as the reference database because we obtained a higher number of mapped reads for this database than in the Refseq and NT databases (Appendix A). Moreover, although the EBCs datasets presented “ancient” taxa, the number of reads mapped to the references did not exceed 20 reads so were not considered in the analysis (Appendix A).

For the damage analysis using ChangePoint, we used collapsed reads of the 29 datasets pre-filtered with the selected pipeline (post-filtered 55 kx) separated in ranges of 50 bp fragment length files (0–50 bp; 51–100 bp; 101–150 bp; 151–200 bp; 201–250 bp; 251–300 bp). Shotgun sequencing results in a pool of DNA fragments of varying sizes; thus, we separated our datasets according to fragment size for this analysis to examine if shorter fragments (<100 bp) showed significantly higher DNA damage compared to longer fragments (>100 bp), as expected for authentic aDNA. First, the proportions of A, T, C and G were generated for both 3′ and 5′ ends, using the fastq files as input. Text files with the proportions were then used to generate the stats of damage profiles in R (dAmIn8r.R). The results were plotted using the script Damage.Analysis.R in R [51]. 

## 3. Results and Discussion

### 3.1. How to Collect and Process Soil Samples for Reproductible Environmental aDNA Studies?

Contaminant DNA is a primary concern when handling aDNA, and the best way to minimize its impact is to take various precautionary measures in each step of sample processing [21,52]. Samples in this case study were processed by taking all the precautions listed below to minimize the contamination of endogenous aDNA. 

#### 3.1.1. Sample Collection

The precautions began with sample collection in the field and we were able to fundamentally reduce exogenous DNA contamination and preserve sample integrity [53]. Llamas et al. [31] proposed guidelines for sample handling in aDNA studies and recommended the following several key precautions: (1) the use of disposable gloves and changing them between samples; (2) avoiding water to wash samples; and (3) the storage of samples in cold (−20 °C to 4 °C) and dry places immediately after collection to avoid freeze/thaw cycles. The authors also recommend the use of protective gear, protection of the sampling site, dedicated trained staff, clean sampling tools, keeping metadata records, in situ samplings of contaminant profiles when allowed, and avoiding chemical treatment to preserve samples [31]. In addition, researchers should collect control samples to monitor contaminant DNA where samples were collected and stored (e.g., air, gloves, working benches, empty sampling tubes). These controls reflect the sampling environment and should be processed similarly to biological samples and considered during the bioinformatic analysis of samples [21,52].

#### 3.1.2. Sample Preparation and Subsampling in aDNA Facilities

To reduce the risk of contamination during laboratory analysis, several publications outlined strict procedures to follow when processing samples for aDNA (reviewed in [31,54,55,56,57,58,59]. Importantly, samples should only be processed in a dedicated aDNA laboratory with the following standards: HEPA-filtered ventilation; positive air-pressure; strict chemical cleaning procedures of surfaces, equipment, and laboratory supplies (using sodium hypochlorite or any DNA degrading detergents and UVC irradiation (>1.45 J·cm^−2^)); strict use of protective gear (i.e., disposable overalls, surgical masks, visor, a minimum of two layers of medical gloves to change the outer layer of gloves regularly, and bleached dedicated footwear); and storage of samples, DNA, and reagents should occur in separate rooms or fridge/freezers. Further, the aDNA facilities should minimally contain physically separated rooms for sample preparation and DNA extraction/library preparation. Importantly, modern samples should not be processed in the aDNA facility, as they represent a high risk of contamination with exogenous modern DNA. Lastly, DNA amplification and sequencing must be performed in a physically isolated post-amplification (i.e., post-PCR) facility.

To remove or minimize pre-laboratory surface contamination, sample decontamination prior to subsampling and DNA extraction is required. However, soil or sediment samples are difficult to decontaminate using classical methodologies (e.g., UV irradiation and sodium hypochlorite) routinely applied to bones, hair, or dental calculus [60,61,62]. Removing the outer surface of a soil or sediment core in an aDNA facility is a more suitable decontamination method [14,63,64]. If using coring equipment, there are several contamination tracing approaches [14]. For example, fluorescent microspheres can be introduced near the coring head during sample collection to simulate particle movement and later ensure that outside signatures of soil or sediment cores are removed before collecting an internal sample for downstream aDNA analysis [65]. 

Environmental microbial communities can have significant spatial and temporal variability at fine geographical scales and are highly heterogeneous [30,66,67,68]. Therefore, researchers should be aware of these biases when subsampling soil or sediment in the aDNA laboratory and take several steps to increase reproducibility, maximize accuracy in reconstructions, and avoid biases during DNA extractions. These procedures include homogenization of the subsample after the removal of the surface, sample randomization, and replication during DNA extraction and sequencing library reparation. Homogenization, or mixing the original sample to obtain a random distribution of all particles, should be performed after surface decontamination by grinding, shearing, beating, or mixing the soil [69]. After homogenization, replicates of each sample type and treatment group should be incorporated to limit batch effects. Lastly, samples should be randomized before DNA extraction to limit the influences of the day-to-day variation of factors, such as contaminant DNA, and further improve sensitivity, rigor, and comparability between samples [70]. 

Finally, laboratory-specific contaminants (i.e., DNA present in laboratory facilities and reagents) should also be monitored by introducing extraction blank controls (EBC). For example, a sterile and empty subsampling tube (e.g., Eppendorf tube) should be opened in the subsampling area for 15–30 s to monitor cross-contamination between samples caused by the dust or aerosol droplets created when opening tubes or transferring samples or liquids [71]. This sample should be included at a minimum ratio of 1 EBC to every 12 samples but can be used more frequently (e.g., as the first and last sample during each extraction) [70]. In addition, the subsampling area should be cleaned thoroughly between samples, using the strict cleaning procedures described above. 

### 3.2. How Are Laboratory Practices and Bioinformatic Workflows Affecting the Reconstruction of Ancient Soil Microbiota?

A successful aDNA extraction method is critical for the accurate reconstruction of a microbial community. Different DNA extraction methods have specific biases and selection for different organisms, especially with HTS techniques [72]. Therefore, the most appropriate method depends largely on the sample type and the study target, although the same DNA extraction methodology is required to compare different samples. Large-scale modern microbiome research projects (e.g., Earth Microbiome Project and Human Microbiome Project) have standardized DNA extraction protocols by using a commercial kit (e.g., DNeasy PowerLyzer PowerSoil kit, Qiagen) [73,74,75]. Commercial kits are a quick and effective way of obtaining environmental DNA while reducing external biases and cross-contamination of samples [75]. However, aDNA is present at low abundances and is fragmented in ancient samples; thus, the DNA extraction method needs to be optimized to reduce DNA loss and efficiently recover short aDNA fragments, which can include specialized silica binding steps [6,34]. Extraction protocols for aDNA have been primarily standardized using human and animal archaeological remains. Only a few available paleomicrobiological studies optimized these methods for the recovery of microbial aDNA, especially in environmental samples. For example, Armbrecht et al. [33] optimized a sedimentary aDNA (sedaDNA) isolation method to increase the extraction efficiency of <500 bp sedaDNA fragments from marine microeukaryotes by comparing extraction treatments with shotgun sequencing of marine sediments collected at Maria Island, Tasmania. The highest rates of aDNA recovery were obtained using a protocol involving bead-beating, EDTA incubation, and a DNA binding step using silica solution in QG buffer [33]. Similarly, Hagan et al. [15] examined microbial aDNA recovery from well-preserved human and dog paleofeces. For this, the authors compared total DNA yield and microbial community structure using five DNA extraction protocols (Dneasy PowerLyzer PowerSoil kit and four variants of an aDNA-optimized modified MinElute protocol for bone extraction). The findings showed that the protocols developed specifically for the recovery of aDNA (i.e., those that used powerbead tubes, QG buffer, PB buffer, and MinElute columns) resulted in significantly higher DNA yields compared to the commercial kit; however, all the protocols showed a consistent taxonomic profile [15]. 

After DNA extraction, the next step was to prepare the sequencing libraries. Currently, the following two DNA-based approaches are typically used to characterize the ancient environmental microbiomes: amplicon sequencing and shotgun metagenomics. Amplicon sequencing, such as 16S rRNA metabarcoding, was used as an early tool in paleomicrobiology [76,77,78,79,80]. However, this technique has several limitations and challenges when used in ancient microbiome studies. For example, the primers used in standard amplicon analyses target regions are limited to certain species and may target genomic regions that are longer than the typical length of aDNA fragments (e.g., <100 bp) [20], potentially favoring modern DNA fragments and increasing the level of contamination compared to the ancient endogenous signal [17]. The whole-genome shotgun metagenomics method provides solutions to these issues, allowing researchers to sequence and identify degraded short ancient DNA fragments from a wide range of species with less amplification bias. Thus, it has become the gold standard for DNA-based paleomicrobiological studies. For example, researchers characterizing ancient dental calculus used shotgun sequences to reconstruct the microbial diversity preserved within a wide range of calculus samples (e.g., [17,77]). 

Resulting sequencing data are processed by a bioinformatic analysis pipeline to reconstruct the soil microbiome. Pre-processing raw HTS sequence reads prior to annotation is a critical step for optimal downstream analysis and comprises the following series of steps: adapter removal, collapse of overlapping paired-end reads, quality/size-filtering, and duplicate removal (reviewed in [81]). As HTS of short DNA fragments may result in the incorporation of adapter sequences into the dataset, contaminating the dataset and negatively impacting the analysis [37], adapter contamination must first be removed. Second, if applicable, collapsing (merging) paired-end reads can be applied. In short-insert libraries, detecting overlapping reads and merging these sequences can reduce the error rate by a factor of five [81]. Next, a quality-score-based filter can be applied to all bases of the read to assure data quality [82], as most HTS technologies immobilize the sequencing templates for bridge-amplification, creating randomly scattered clusters that can cause mixed-signal readouts [83]. Lastly, other filters, such as low-complexity filtration and deduplication, are used to eliminate other HTS biases. Low-complexity sequences are removed from the dataset because they are usually caused by sequencing artifacts [38]. Additionally, the removal of duplicated reads (deduplication) is performed under the assumption that PCR amplification is responsible for most of the read duplications in HTS data [84]. In summary, data pre-processing is of particular importance when working with aDNA containing post-mortem DNA modifications towards the end of the read, which could cause high error rates during sequencing. However, data preprocessing usually requires a combination of multiple conventional tools to perform the quality control (e.g., FastQC), adapter removal (e.g., AdapterRemoval2 or Cutadapt [85]) and filtering (e.g., Trimmomatic [86]), making this process slow and inefficient. New tools are being developed to facilitate this process, such as the ultra-fast all-in-one tool Fastp, which incorporates quality control and data-filtering features [38].

After pre-processing, the next step is to identify the microbial taxa present in ancient samples. Previous studies have benchmarked metagenome taxonomic classifiers for ancient microbiome research and determined that alignment-based approaches are minimally affected by aDNA deamination, compared to assembly based and alignment-free classifications [18,19]. Of the currently available programs, the alignment-based software MALT (MEGAN Alignment Tool) has been shown to outperform the alignment of short, fragmented DNA than some other programs (e.g., [17]). Eisenhofer and Weyrich [19] corroborated this finding and observed that nucleotide-to-nucleotide alignments were improved over nucleotide-to-protein (e.g., MALTn to MALTx). However, MALT is computationally intensive and therefore is likely to be out of the resource’s capacity of many researchers. Further, other metagenomic classifiers have shown good performance with short reads in aDNA studies, such as Kraken2 and Centrifuge [87,88,89].

Several research teams have also explored the impact of different reference databases and found that database choice can significantly bias the results of alignment-based taxonomy classification in ancient metagenomic studies of human-associated microbes [19,90]. However, this effect is likely to be much higher in environmental studies that have fewer reference sequences. Nevertheless, previously utilized databases for ancient microbiome analyses include SILVA SSU 132 [44], NCBI nt database [91], NCBI RefSeq [91], and HOMD [92] (e.g., ancient microbiome studies: [17,19,33,93,94]). 

Here, we integrated these concepts into the case study described below to examine methodological, technical, and analytical biases during an environmental aDNA study of prokaryotes. 

### 3.3. Benchmarking Laboratory Methods and Bioinformatic Analysis for the Recovery of Microbial Ancient DNA from Soil Samples 

#### 3.3.1. Impact of Extraction Methods and Data Preprocessing Pipelines on Read Number and Fragment Length Recovery

In our case study, after de-multiplexing, adapter-trimming, and collapsing paired-end sequences in the 29 datasets, we retained a range of 28,586–37.4 M collapsed reads per dataset for soil samples, and a range of 0–184,078 collapsed reads per dataset for EBCs (pre-filtered, Appendix A). After filtering low-complexity and duplicated sequences, we retained 5685–30.6 M collapsed reads per dataset for samples and 0–79,983 collapsed reads per dataset for EBCs (post-filtered; Appendix A). MultiQC revealed that >99% of the sequences in pre-filtered and post-filtered sample datasets had a quality score of >Q30 (Appendix A). The collapsed read length average ranged from 44 to 178 bp for samples and 48 to 126 bp for EBCs datasets (Figure 2 and Appendix A). The GC content for collapsed reads averaged from 47% to 65% for samples and 47% to 57% for EBCs (Appendix A). The comparison of both preprocessing tools (i.e., AdapterRemoval2 and Fastp) resulted in non-significant differences in the number of collapsed reads (Tukey HSD test, *p* > 0.05; Appendix A). Significant differences in read length were observed between datasets preprocessed using Fastp and ≤55% low-complexity thresholds and datasets preprocessed with AdapterRemoval and a 70% low-complexity threshold (Tukey HSD test, *p* < 0.05; Appendix A). An increasing threshold of low-complexity filters showed a significantly higher retention of shorter fragments (Tukey HSD test, *p* < 0.05; Appendix A). As expected, pre-filtered duplication levels were significantly higher than post-filtered duplication levels (Tukey HSD test, *p* < 0.05; Appendix A) and did not show an impact on the average length of sequences (Appendix A). In general, the most significant effects on sequence pre-processing was observed when we increased the low-complexity filtration threshold, while other steps had limited impacts on the data.

Overall, the SiO_2_ + PB buffer DNA extraction method resulted in the highest number of short, collapsed reads in three of the four samples, with the exception being Sample 4. Sample 4 had a higher number of short collapsed reads using the SiO_2_ + QG buffer method (Figure 2; Appendix A). Specialized silica binding protocols, such as SiO_2_ + PB buffer and SiO_2_ + QG buffer methods, have been widely used to efficiently recover prokaryotic and eukaryotic fragmented and degraded aDNA fragments from sediments and other sample types [6,9,17,33,95]. The SiO_2_ + PB buffer method was reported to have better recovery of small DNA fragments than the SiO_2_ + QG buffer method [6]; our results are consistent with those findings, despite the results of Sample 4. The site where Sample 4 was collected corresponds to a former paleolake covered by a layer of sandy clay [96], making the recovery of extracellular DNA, and specifically of shorter fragments, particularly challenging due to the strong capacity of clay to absorb and bind DNA molecules [97,98]. In Sample 4, it is possible that the use of Guanidine thiocyanate (QG buffer)—a stronger chaotropic salt compared to guanidine hydrochloride (PB buffer)—could have facilitated a better overall recovery of shorter fragments compared to other protocols. Similar results were obtained by Zainabadi et al. [99], who observed that guanidine thiocyanate outperformed guanidine hydrochloride during the purification of small-sized nucleic acid molecules from human urine.

The SiO_2_ + PowerLyzer, DNeasy PowerLyzer PowerSoil kit, and phenol–chloroform methods recovered longer fragments across all samples (Figure 2; Appendix A). While DNA extraction kits offer rapid sample processing and a high-resolution of paleo-community data [33], the cell lysis and inhibitor removal steps of these kits resulted in DNA losses [15], especially of shorter DNA fragments, and selectively retained long DNA fragments. While these kits may be appropriate for modern microbial DNA studies of soil and sediment, their use in ancient environmental studies should be approached with caution. 

#### 3.3.2. Effect of Contaminant Filtering

In general, fewer DNA sequences were found using the SiO_2_ + PB buffer and SiO_2_ + PowerLyzer methods, respectively (Appendix A) in EBCs 1 (presence of contaminants in reagents and environment), followed by the DNeasy PowerLyzer PowerSoil kit, phenol–chloroform method, and SiO_2_ + QG buffer (Appendix A). In EBC2s (presence of cross-contamination), no DNA sequences were recovered for the phenol–chloroform method, and fewer DNA sequences were found when using the SiO_2_ + Power Lyzer and SiO_2_ + PB buffer methods, followed by the Dneasy PowerLyzer PowerSoil kit and SiO_2_ + QG buffer (Appendix A). More DNA sequences were obtained in EBC1s than in EBC2s (20% to 100% decrease) in SiO_2_ + PowerLyzer, SiO_2_ + PB and phenol–chloroform, indicating a lower contribution of cross-contamination during ancient eDNA analysis using these methods (Appendix A). On the contrary, SiO_2_ + QG and Dneasy PowerLyzer PowerSoil kit presented a higher number of reads in EBC2s than EBC1s (50 to 207% increase) (Appendix A). Further, DNA sequences were markedly reduced post-filtering for EBC1s (37% to 99% decrease) and EBC2s (32% to 98% decrease) (Appendix A). While these findings suggest that minor contaminant species’ DNA was recovered from the SiO_2_ + PB method, contaminant species were recovered from all methods, highlighting the importance of including EBCs in laboratory analyses to monitor contaminant DNA. Overall, DNA extraction with the SiO_2_ + PB buffer method presented higher retention of shorter fragments indicative of ancient sequences and contained fewer contaminant sequences. 

Contaminant species can significantly inflate or alter signals within microbiome datasets, hindering our ability to characterize microbial communities accurately [47,70]. For this reason, contaminant species must be filtered from datasets before analyzing the taxonomic composition and diversity of a sample. Here, we assessed the presence of taxa before and after filtering contaminant taxa of the 160 sample datasets (analyzed with the eight bioinformatic pipelines) to determine the impact of contaminant species removal from ancient soil samples. We also examined and compare the contamination levels in each extraction method.

Before decontamination, 337,829 collapsed reads (12.77% of total 2,645,807 collapsed reads, Appendix A) were taxonomically classified to specific species across samples: 3.70% to Archaea, 54.05% were assigned to Bacteria; and 42.24% to Eukaryota (Appendix A). A high proportion of unassigned reads is expected in ancient metagenomic studies [19], which is likely due to unknown species, especially in poorly studied samples, such as ancient environments. In collapsed reads from EBCs, we detected 17 bacterial and 7 eukaryotic taxa at the species level (Appendix A), which were previously identified as laboratory and reagent contaminants in aDNA and low biomass microbial studies (e.g., Acinetobacter, Enterococcus, Pseudomonas) [33,59,70,100]. The eukaryote, Thecofilosea, was the most abundant contaminant, especially when using the SiO_2_ + PB buffer and phenol–chloroform methods. Similarly, Armbrecht et al. [33] detected Thecofilosea within the phylum Cercozoa as the primary eukaryotic contaminant group; as both studies were completed in the same facilities, the sequences were probably recovered from a shared laboratory contaminant, highlighting the importance of stringent contamination reduction, data filtering, and contaminant monitoring during ancient microbiome studies [13,101]. Moreover, non-collapsed reads from EBCs showed higher proportions of Thecofilosea and other known contaminant sequences than collapsed reads (Appendix A), suggesting higher contamination levels amongst longer DNA fragments in these data. Nevertheless, the majority of sequences identified as Thecofilosea were short (<50 bp) and presented nucleotide repeats (i.e., low-complexity), which often correspond to sequencing artifacts and results in an ambiguous alignment to multiple locations in a reference sequence, especially in the presence of highly contaminated and low-quality genomic references [102]. These species were also present in the sample datasets and were significantly more abundant in pre-decontaminated samples than in post-decontaminated samples (ANCOM; W> 360; Appendix A). 

After filtering taxa found in EBCs from biological samples, 337,829 collapsed reads were taxonomically classified in the following three domains: 5.43% to Archaea; 43.64% were assigned to Bacteria; and 50,92% to Eukaryota using the SILVA SSU 132 database (Appendix A). To evaluate whether the removal of contaminant species alters microbial diversity, we compared the alpha diversity (observed features and Shannon’s diversity indices) of pre- and post-decontaminated samples, rarefied at 1000 sequences. As expected, every decontaminated sample resulted in lower observed features compared to contaminated samples (Appendix A). However, only Sample 2 showed significant differences between pre- and post-decontaminated samples in their observed features and Shannon’s diversity indices (Kruskal–Wallis; *p*-value = 0.046 and *p*-value = 0.0003, respectively; Appendix A). Further, a PCoA based on Bray–Curtis and Jaccard dissimilarity showed that post-decontaminated datasets (Appendix A, respectively) presented clearer clustering according to sample type compared to pre-decontaminated dataset (Appendix A, respectively), suggesting that shared contaminant signatures dampened sample-specific beta-diversity signatures. Moreover, the composition of microbial communities in pre-decontaminated datasets was significantly different from that in post-contaminated datasets across all samples (Bray–Curtis PERMANOVA, *t* = 13.99; *p* = 0.001; Appendix A). Overall, contaminant filtering appears to reduce noise, with minimal impacts on endogenous diversity within ancient soil datasets.

#### 3.3.3. Taxonomic Biases of DNA Extractions Methods

DNA extraction protocols can greatly influence the signals obtained in ancient studies due to the low concentration of aDNA in the sample compared to modern DNA, varied resilience of taxa to the cell-lysis method, and DNA binding capacities of different soil and sediment types [6,15,33,95]. After removing EBCs taxa and singletons (read with a sequence present once in the data) from the communities identified using the SILVA SSU 132 database, we compared the taxonomic profiles, alpha diversity, and beta diversity of rarefied (1000 sequences at species level) datasets obtained using the five different extraction methods. A comparison of the phyla within each sample was considerably different between DNA extraction protocols across all three domains of life (Figure 3, Figure 4 and Figure 5). Overall, abundant phyla (e.g., Acidobacteria, Chloroflexi, Crenarchaeota, Firmicutes, Proteobacteria, Streptophyta; Appendix A) were present across all datasets. However, differences in taxonomic profiles were larger in domains with fewer assigned reads (i.e., Archaea), rare taxa with lower abundances, and extraction methods with overall poor DNA yield (e.g., those using phenol–chloroform extraction method) (Figure 3, Figure 4 and Figure 5). For example, reads assigned to Archaea could only be recovered with SiO_2_ + QG buffer in Sample 3 and with the PowerLyzer kit and SiO_2_ + QG buffer in Sample 4 (Figure 3), and no bacterial or eukaryotic sequences were recovered when using the phenol–chloroform method in Samples 3 (Figure 4 and Figure 5), likely due to the poor recovery of DNA fragments. Overall, the extraction method impacted the taxa recovered from each sample. 

We also examined this observation across non-collapsed reads, as these sequences may be longer and reflect modern microbial communities living in these environments. The composition of microbial communities in non-collapsed sequences was significantly different from that in collapsed reads across all samples (PERMANOVA, *t* = 11.33; *p* = 0.001; Appendix A). Species, such as *Streptococcus pneumoniae*, unclassified Endomicrobiaceae, unclassified Thiothrix, and *Pseudomonas putida*, were significantly higher in non-collapsed sequences compared to collapsed reads (ANCOM; W > 429; Appendix A). Further, non-collapsed reads of samples also showed a high proportion of reads assigned to Thecofilosea, the taxa that was highly abundant in EBCs and is likely a modern contaminant (Appendix A). This suggests that a more significant number of contaminant or modern sequences may more often present in the non-collapsed sequences, as expected [17]. While overall diversity in this dataset is more tightly linked to sample type, small scale biases in diversity and composition due to extraction methods are still evident. For example, alpha diversity (observed features and Shannon’s diversity indices) slightly decreased in samples extracted with methods with poor DNA yields compared to samples with a higher number of collapsed reads (Appendix A). Significant differences were also found in observed features index between extraction protocols (Kruskal–Wallis; *p*-value < 0.05; Appendix A). Further, the composition of microbial communities between different extraction protocols was significantly different across all samples (PERMANOVA, *t* = 2.36; *p* = 0.004; Appendix A). The PCoA of Bray–Curtis and Jaccard diversity metrics for beta-diversity showed that the clustering of reconstructed microbial communities is primarily driven by the signatures present in the biological samples, but the extraction method also impacts composition (Appendix A). Hagan et al. [15], showed that ancient microbiota of animal-related samples clustered primarily by sample source, rather than extraction method. While the previous study was limited to animal-related samples [15], here we showed that DNA signals recovered from environmental samples are potentially more susceptible to contamination introduced by modern microbes living in these environments and laboratory protocols. These findings may have implications for the meta-analysis of published datasets that employ different methodologies.

Overall, we recommend using the SiO_2_ + PB buffer and SiO_2_ + QG buffer methods to recover short microbial DNA sequences from ancient soil samples, according to their performance in this case study and previous aDNA studies (e.g., [6]).

#### 3.3.4. Bioinformatic Preprocessing Has Minimal Impacts on Overall Prokaryotic Taxonomic Profiles

We examined the impact of different bioinformatic strategies on microbial composition and diversity using taxa identified with the SILVA SSU 132 database. Although ribosomal RNA (rRNA) genes represent a small proportion of genomes, we selected SILVA SSU 132 as it is a comprehensive curated reference database that allowed us to taxonomically classify our 232 datasets in the three domains of life with low memory consumption. Overall, we assessed the impact of preprocessing software (AdapterRemoval2 vs. Fastp), deduplication, and low complexity filtering on species recovery. Non-significant differences in alpha diversity (observed features and Shannon’s diversity indices) (Kruskal–Wallis; *p*-values > 0.05, Appendix A) and beta diversity (PERMANOVA, *t* = 1.89; *p* = 0.057; Appendix A) were observed between collapsed reads preprocesses with the software AdapterRemoval2 + Komplexity + Dedupe and FastP + Dedupe. 

On the other hand, FastP marginally improved the annotation of eukaryotic species, as additional Eukaryotic taxa (e.g., species belonging to Platyhelminthes phylum) were present in the Fastp analyzed samples but not in profiles analyzed using AdapterRemoval2 (Figure 5; Appendix A). Because our analyses focused on assessing the impact of bioinformatic pipelines on prokaryotes, we did not further explore this effect; nevertheless, it should be the subject of future research. Furthermore, duplication levels and adapter content were lower in datasets preprocessed with bioinformatic pipelines using AdapterRemoval2 + Komplexity + Dedupe (Appendix A). For low complexity filtering, a 70% threshold resulted in a lower recovery of reads per taxa at the species level compared to lower thresholds, as expected (Appendix A). However, non-significant differences in alpha diversity (Kruskal–Wallis; *p*-values > 0.05; Appendix A) and beta diversity (PERMANOVA, *t* = 0.811; *p* = 0.63; Appendix A) were observed on species recovery between datasets analyzed with 30%, 55% or 70% low-complexity thresholds. Bray–Curtis and Jaccard PCoA results show that the clustering of reconstructed microbial communities is again primarily driven by the signatures present in the biological sample and not low-complexity filtering (Appendix A). Samples scattered in the center of the PCoA plots (red circles, Appendix A) correspond to pre-filtered samples, which contain low-complexity reads and high levels of read duplication (up to 95%) that could mask the microbial signal and significantly interfere with the interpretation of results. This effect was completely removed in all the post-filtering datasets (Appendix A), where we observed the distinct clustering of datasets driven mainly by the sample of origin and DNA extraction protocol. We selected a low complexity threshold of 55% for all comparisons, as it represented an intermediate value that was not too permissive or restrictive, taking into account that prokaryotic genomes naturally present short-sequence DNA repeats (SSR) [103]. For deduplication, removing only exact duplicate sequences versus removing exact sequences plus sequences with two mismatches showed non-significant differences in alpha diversity (Kruskal–Wallis; *p*-values > 0.05; Appendix A) and beta diversity (PERMANOVA, *t* = 0.853, *p*-value = 0.584; Appendix A). Moreover, the Bray–Curtis and Jaccard PCoA results showed that the clustering of reconstructed microbial communities is again primarily driven by the signatures present in the biological sample and extraction method, and not the deduplication method (Appendix A).

Overall, we selected the software AdapterRemoval2 to demultiplex, adapter trim and collapse our sequences, complemented with the use of Komplexity (55%) to remove low complexity reads and Dedupe to deduplicate the data for downstream analysis.

#### 3.3.5. Impact of Reference Database Selection on Species Recovery

We next examined the impact of reference databases on the reconstruction of microbial communities by comparing the collapsed, preprocessed, and decontaminated sequences from our selected pipeline (post-filtering_55%) and the five extraction methods using the following four databases: SILVA SSU 132; RefSeq, NCBI, and GTDB. However, for the diversity analysis comparison, we removed SILVA datasets as they contained a low numbers of sequences compared to the rest of the data. Further, the comparison of taxonomic profiles and diversity analyses between databases were performed using collapsed sequences assigned to genera within the Bacteria domain, as we obtained a higher number of reads and taxonomic classification compared to reads assigned to the Archaea domain. 

The highest number of classified genera was found using the SILVA SSU 132 database, followed by the GTDB, Refseq, and NT databases, respectively (Appendix A). However, approximately 50% of the classified sequences remained unclassified, showing low taxonomic resolution at deeper ranks when using SILVA SSU 132 and GTDB (represented in black in Appendix A). Further, bacterial composition from two samples (i.e., Sample 1 extracted with SiO_2_ + PowerLyzer kit and Sample 3 extracted with phenol–chloroform) could not be reconstructed when using Refseq and NT databases, as sequences could not be successfully classified with the selected downstream LCA parameters in MEGAN6 (i.e., 90% identity, 5% minimum support, weighted LCA algorithm) (Appendix A).

Alpha diversity indices (observed features and Shannon’s diversity) were higher with GTDB databases (Appendix A), followed by Refseq and the NT databases, respectively (Appendix A). Significant differences were observed in alpha (Kruskal–Wallis; *p*-values > 0.05; Appendix A) and beta diversity (PERMANOVA, *t* = 0.853, *p*-value = 0.584; Appendix A) between databases, except between the Refseq and NCBI databases. Previous studies observed significant biases associated with databases in the reconstruction of ancient metagenomes [17,18,19]. However, our results suggest such biases may be exacerbated in environmental samples, especially those from less-studied ecological niches such as ancient sediments from extreme environments; this further emphasizes the importance of choosing a suitable database in the metagenomic analysis of ancient samples. Here, we found that certain databases, such as GTDB, may be better suited to providing a higher taxonomic classification for ancient environmental samples than human-associated samples, likely due to the collections of environmental species included in the reference database.

### 3.4. Environmental DNA Pool: Authenticating the aDNA Signal 

Despite the rapid growth in paleomicrobiology with HTS, a considerable number of publications lack robust aDNA authentication and have been widely criticized (e.g., [70]). Authentication tools developed for paleogenomics rely on identifying fragment lengths and damage patterns consistent with DNA decay. Briggs et al. [104] suggested that nucleotide misincorporation patterns of deaminated cytosines (uracils) in single-strand overhangs formed post-mortem underpin a powerful approach to authenticate aDNA generated using HTS. Later, Sawyer et al. [105] determined that the misincorporation proportions correlated with the sample’s age. These ‘damaged patterns’ in the DNA now serve as an unequivocal signature of authentic ancient DNA molecules. The application of these tools has indeed verified that microbial DNA degrades at appreciable rates, although they may be different across species with different G/C ratios or slower than the rates observed in humans [42,106]. In addition, typical genetic variation of damaged reads can also be used to validate ancient sequences using phylogenetic analysis, as ancient genetic variation should fall outside of that seen today [31,54]. However, these authentication tools were developed to identify a single species of interest (e.g., humans) and not a complex dataset with a high risk of contamination, such as environmental microbial communities. Although new tools are being developed to process this type of data (ChangePoint [51]; HOPS (Heuristic Operation for Pathogen Screening) [50]; PyDamage [107]; DamageProfiler [108]), damage patterns still need to be better characterized for microorganisms, especially with the relative rates of damage in different species and environments.

For this case study, we evaluated the authenticity of the results across different DNA extraction and bioinformatics protocols by testing the DNA damage patterns using two statistical models: HOPS [50] and Changepoint [51]. HOPS, a tool for high-throughput screening of DNA, calculates the probability of sequence mismatches between the collapsed reads and a reference genome used to align the sequences from the start to the end of reads in metagenomic data. Moreover, a HOPS analysis can be performed using either the default or ancient mode, which screens the data for ancient DNA sequences using all mapped reads, or only reads with damage (i.e., C to T and G to A substitution at 5′ and 3′ end) [50], respectively. However, reference genome biases can impacted alignment-dependent authentication tools, particularly when working with rare and poorly studied taxa, such as those found in extreme environments. The lack of reference genomes for many environmental species represents a significant gap in public repositories and may cause mapping errors during DNA damage analyses. To address these issues, we also included a second authentication model (Changepoint) based on an alignment- free algorithm that does not require a reference genome [51]. 

To compare the performance of the protocols in each sample, we compare DNA damage and edit distance plots of “ancient” taxa of the most abundant species in each dataset (Figure 6). Overall, the HOPS analysis reveals that samples extracted with silica-based methods (i.e., SiO_2_ + PB buffer, SiO_2_ + QG buffer and SiO_2_ + PowerLyzer) contained more damaged DNA sequences (presence of deamination at the 5′ and 3′ ends), that mapped to reference genomes compared to the other protocols (Appendix A; Figure 6). The changepoint analysis also showed that samples extracted with SiO_2_ + PB buffer and SiO_2_ + QG buffer methods contained a significant DNA damage signal at both 5′ and 3′ ends, especially in the shortest fragments (<100 bp) (Figure 7), whereas the other methods did not. Although most DNA damage plots have a continuous decline of C to T and G to A mismatch compared to modern species, their profiles also show signs of mapping to an incorrect representative reference genome (i.e., Figure 6D, *Metallibacterium scheffleri*). Mapping to an incorrect species generally results in increased mismatches across the whole read, which is evident in the edit distance distribution [50]. Finally, EBC samples contained less than 20 damaged reads per taxa, so they were not considered in the analysis (Appendix A). 

To further examine the recovery of ancient DNA molecules from each extraction method, we also compared the damage of individual taxa in each sample. In Sample 1, we compared damage in Acidobacteria bacterium (the only species represented across all datasets of the same sample) (Figure 8, Appendix A). A higher number of damaged reads (727 reads) were observed when using the SiO_2_ + PB buffer than for any other protocol (of 23–622 reads; Figure 7; Appendix A). Across other species in Samples 1–3, SiO_2_ + PB buffer consistently resulted in a higher proportion of damaged reads per taxa across datasets and had corresponding edit distance plots with a continuous decline of mismatch compared to modern reference genomes [34]. However, this was not the case for Sample 4, which had a higher number of damaged reads when the sample was extracted with the SiO_2_ + QG buffer method (Appendix A; Figure 6). Furthermore, all taxa obtained using SiO_2_ + PB buffer and SiO_2_ + QG buffer protocols with >1000 damaged reads had DNA damage profiles and edit distance plots with a consistent decline of mismatch, consistent with previous observations [16] and the presence of aDNA in our data. In summary, the results obtained with the software HOPS and Changepoint were consistent. In addition, our results suggest that authentic aDNA signals can be consistently obtained when using either the SiO_2_ + PB buffer and SiO_2_ + QG buffer DNA extraction methodologies. 

## 4. Conclusions

Ancient DNA research is a powerful new tool used to reconstruct the evolutionary history of environmental microbes. Ancient DNA has already helped to describe how microbial communities adapt to environmental factors and shape complex ecological processes over time. Although ancient environmental microbiome studies have largely proliferated in the last decade, this is the first systematic study conducted to assess the performance and potential biases of different DNA extraction protocols and bioinformatic strategies on the recovery of microbial aDNA from terrestrial soil samples. We found that silica-based DNA extraction protocols optimized to obtain aDNA, mainly SiO_2_ + PB buffer and SiO_2_ + QG buffer, showed better performance in recovering short fragments (<100 bp) with authentic aDNA signal compared to commercial kits and the phenol–chloroform method. We also demonstrated that reducing low-complexity and duplicated reads, as well as removing taxa commonly identified as modern DNA contaminants, can reduce noise in ancient soil datasets. Our results corroborate biases introduced from database selection and identified SILVA SSU 132 and GTDB as effective databases to recover ancient environmental species. The guidelines reviewed and proposed in this paper will contribute to and facilitate the development of future ancient soil/sediment microbiome studies, and future research should examine the cellular degradation of microbes and whole microbial communities in different environmental contexts.

## Figures and Tables

**Figure 1 microorganisms-10-01623-f001:**
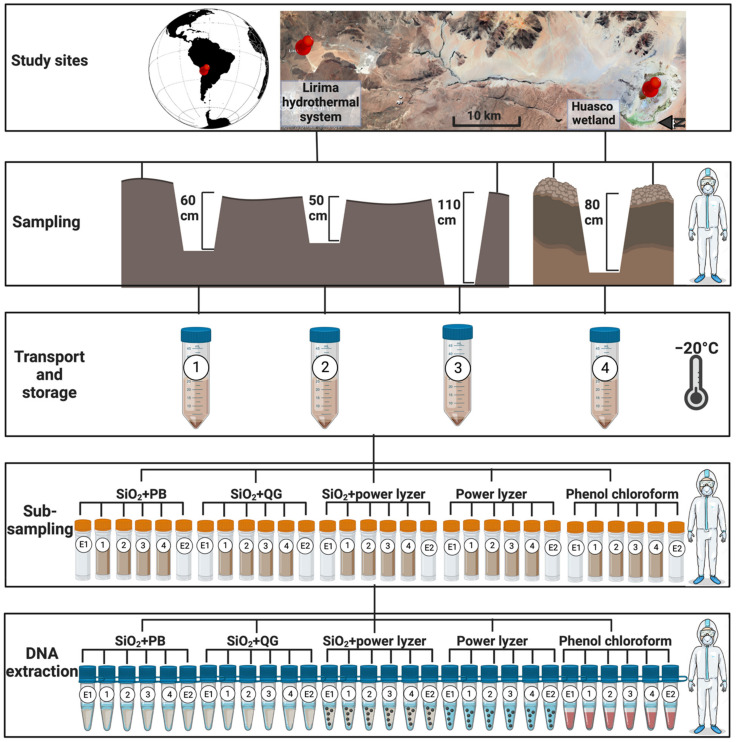
Diagram depicting procedures from sampling collection to DNA extraction. E1: Extraction blank control 1 (EBC1); 1: Sample 1; 2: Sample 2; 3: Sample 3; 4: Sample 4; and E2: Extraction blank control 2 (EBC2). Study sites: Satellite image with the location of Lirima hydrothermal system and Huasco wetland marked. Sampling: Sample 1 was taken at 60 cm depth, Sample 2 at 50 cm, Sample 3 at 110 cm and Sample 4 at 80 cm. Transport and storage: samples were transferred from study site to the lab in liquid nitrogen and then storage at −20 °C. Sub-sampling: The four samples were sub-sampled in a specialized aDNA facility, and two extraction blanks were created for every extraction batch. DNA extraction: sub-samples and extraction blanks were extracted using five different extraction methods (SiO_2_ + PB, SiO_2_ + QG, SiO_2_ + PowerLyzer, PowerLyzer and phenol–chloroform).

**Figure 2 microorganisms-10-01623-f002:**
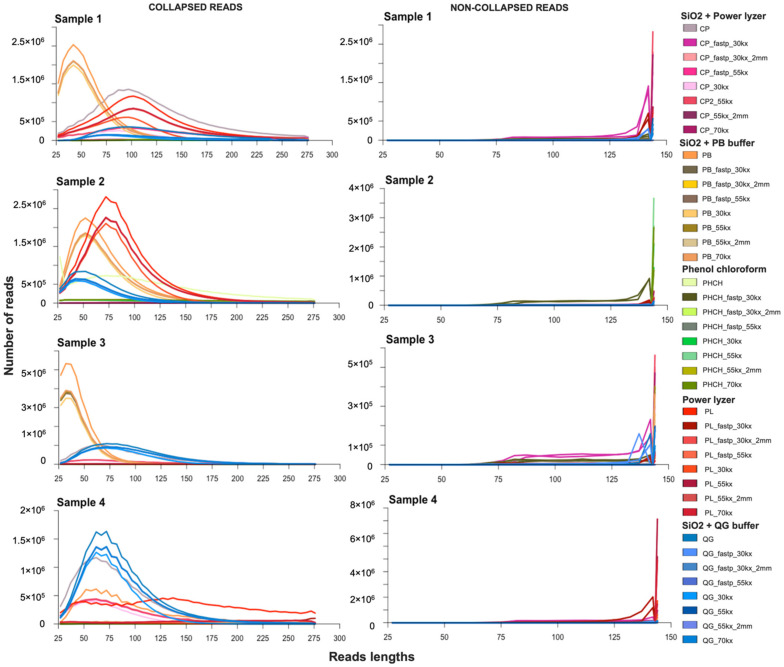
DNA fragment length distributions of collapsed (**left**) and non-collapsed (**right**) reads from the four samples extracted with five DNA extraction methods and preprocessed using eight bioinformatic pipelines. Each color indicates a different extraction method and pipeline.

**Figure 3 microorganisms-10-01623-f003:**
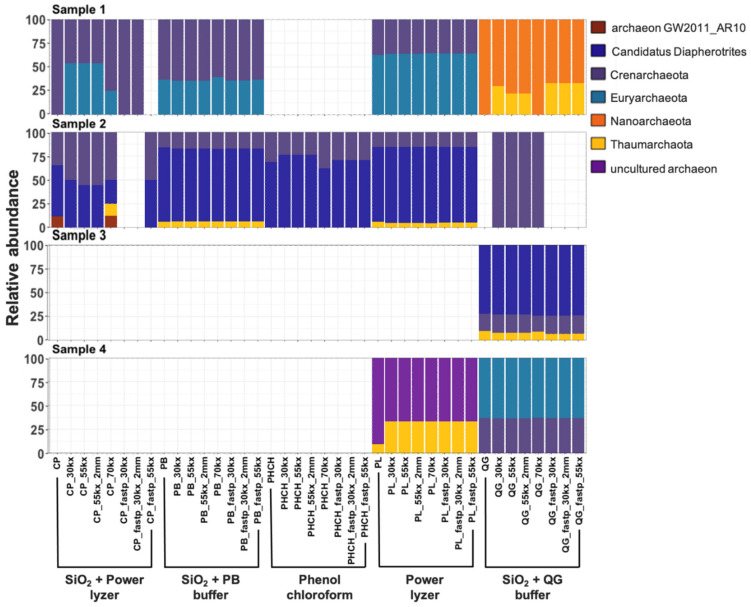
Taxonomic profile for the Archaea domain at phylum level of collapsed reads of the four samples extracted with five DNA extraction methods and preprocessed using eight bioinformatic pipelines, obtained by aligning collapsed reads to SILVA SSU 132 as reference database.

**Figure 4 microorganisms-10-01623-f004:**
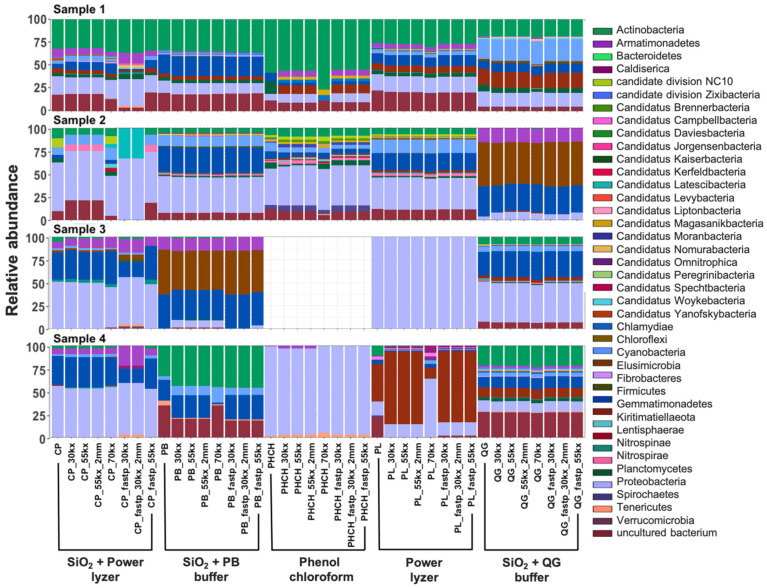
Taxonomic profile for the Bacteria domain at phylum level of collapsed reads of the four samples extracted with five DNA extraction methods and preprocessed using eight bioinformatic pipelines, obtained by aligning collapsed reads to SILVA SSU 132 as reference database.

**Figure 5 microorganisms-10-01623-f005:**
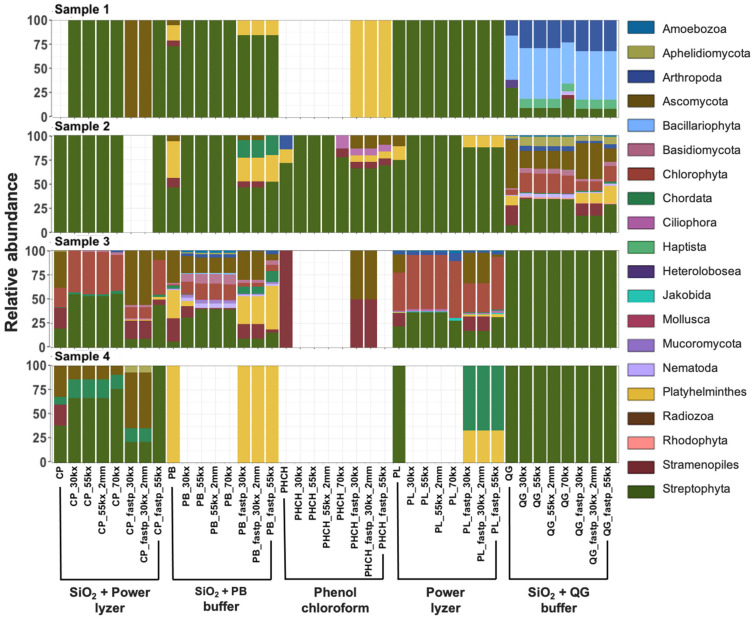
Taxonomic profile for the Eukaryota domain at phylum level of collapsed reads of the four samples extracted with five DNA extraction methods and preprocessed using eight bioinformatic pipelines, obtained by aligning collapsed reads to SILVA SSU 132 as reference database.

**Figure 6 microorganisms-10-01623-f006:**
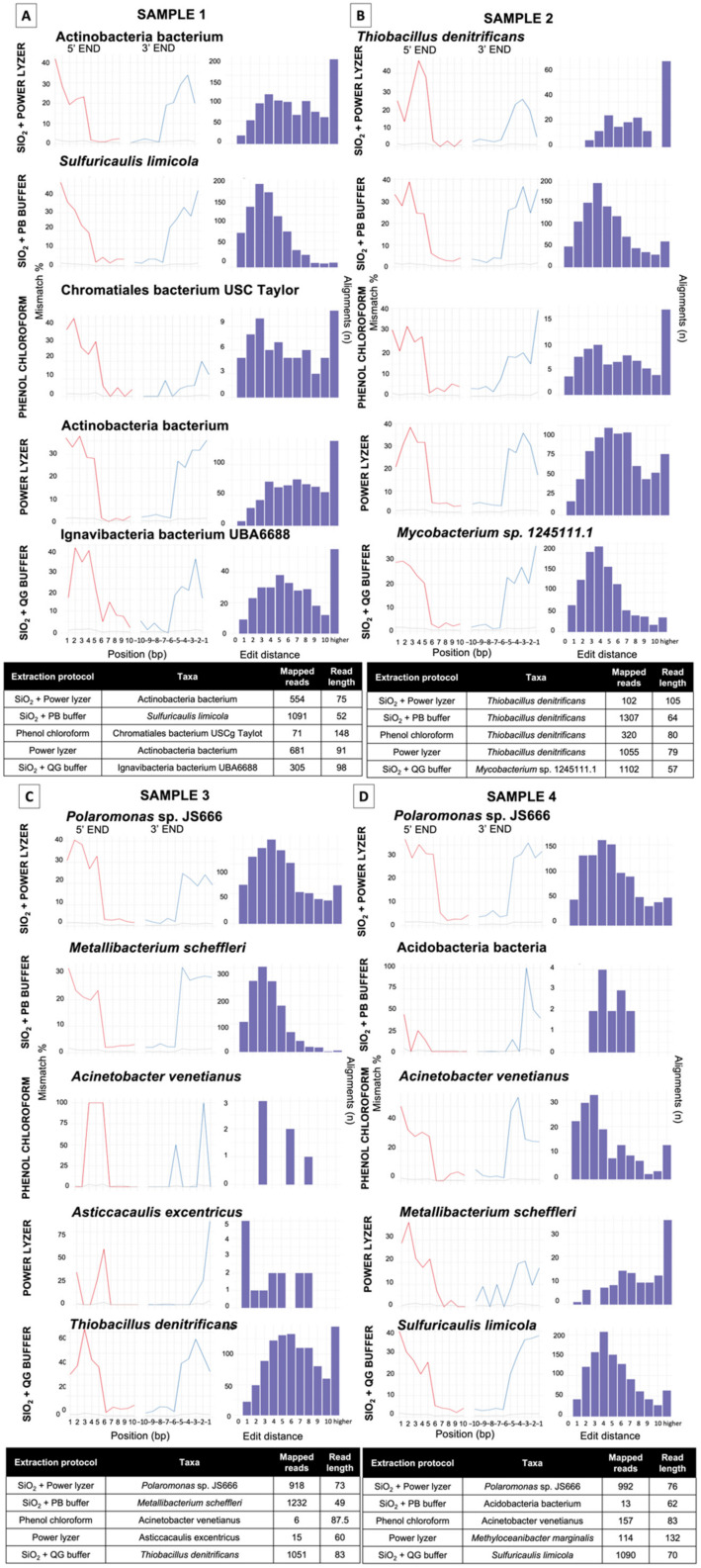
DNA damage profiles of the most abundant species per dataset. Cytosine deamination and edit distance plots of the most abundant species in (**A**) Sample 1, (**B**) Sample 2, (**C**) Sample 3 and (**D**) Sample 4, extracted using five DNA extraction methods and preprocessed with the selected bioinformatic pipeline (AdapterRemoval v2, 55kx, deduplication of exact sequences). Plot were obtained using HOPS with the ancient filter and the GTDB a reference database. The left and right panels of cytosine deamination plots for each sample display the 5′ C-to-T (red lines) and 3′ G-to-A (blue lines) substitution rates, respectively.

**Figure 7 microorganisms-10-01623-f007:**
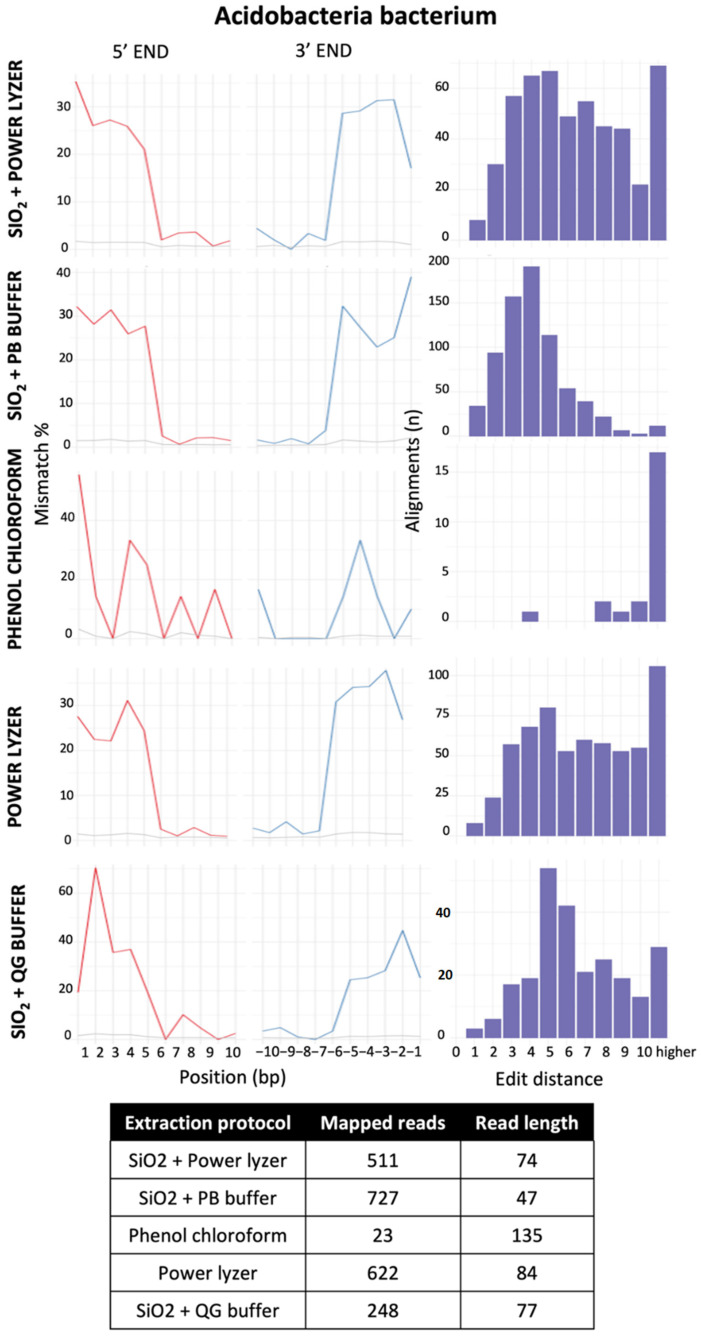
Comparison of cytosine deamination and edit distances plots of reads mapping to the taxa Acidobacteria bacterium across all extraction protocols in sample 1, obtained using HOPS with the ancient filter and the GTDB a reference database. The left and right panels of cytosine deamination plots for each sample display the 5′ C-to-T (red lines) and 3′ G-to-A (blue lines) substitution rates, respectively.

**Figure 8 microorganisms-10-01623-f008:**
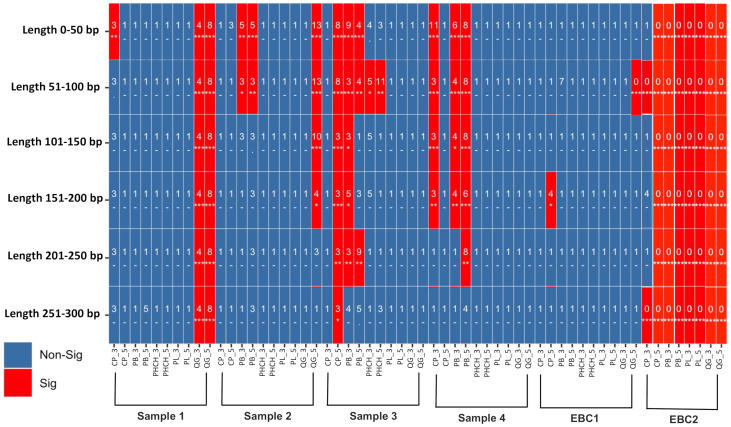
DNA damage profiles of samples 1–4 and EBCs calculated using the alignment-free model ChangePoint. Datasets extracted with five DNA extraction methods and preprocessed with the selected bioinformatic pipeline (AdapterRemoval v2, 55kx, deduplication of exact sequences), were separated in ranges of 50 bp fragment length (0–50 bp; 51–100 bp; 101–150 bp; 151–200 bp; 201–250 bp; 251–300 bp) for this analysis. Red color indicates statistical significance (*p* < 0.050), and number zero indicates no damage.

## Data Availability

All raw sequence data from this study are available at NCBI SRA (https://www.ncbi.nlm.nih.gov/sra, accessed on 5 January 2022) under project number: PRJNA794308. The proportion results generated with ChangePoint and the full list of QIIME2 and R scripts used in the analysis are detailed in https://github.com/VilmaPerez/Soil_ancient_metagenomics, accessed on 5 January 2022.

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
