# Peer review of "A Case Study for the Recovery of Authentic Microbial Ancient DNA from Soil Samples"

_microorganisms, 2022, doi:10.3390/microorganisms10081623_

Round 1

Reviewer 1 Report

Ancient DNA research is a powerful new tool reconstruct the evolutionary history of environmental microbes. Application of aDNA sequencing to better understand bacterial genome evolution and adaptation. However, most aDNA studies have not yet optimized their DNA extraction protocols for microbial aDNA and bioinformatics protocols. Here a a comprehensive study is carried out on aDNA extraction and bioinformatic analysis as well as databases selection. The experimental design is reasonable, and the findings are fascinating. Here are some minor comments:

Line 751: Figure 3, please check Bacteria

Line 755: Figure 4, please check Archaea

Author Response

Reply to reviewer 1: We appreciate the time and effort devoted by the reviewer to evaluate our manuscript. Thank you for pointing out the errors in the legends of Figure 3 (Line 751) and Figure 4 (Line 755). We have fixed these accordingly, and we have tracked changes within the revised manuscript to highlight these amendments.

Reviewer 2 Report

Title: A case study for the recovery of authentic microbial ancient DNA from soil samples

Reference: microorganisms-1795273

Authors: Vilma Pérez, Yichen Liu, Martha Hengst, Laura S. Weyrich

Article type: Review

Reviewer Comments:

The manuscript microorganisms-1795273, entitled “A case study for the recovery of authentic microbial ancient DNA from soil samples”, revises the literature available for environmental ancient microbial projects, comprising all methodological steps from sample collection to data analysis, while it also evaluates previous methodologies and approaches used in published such studies.

General comments:

1. Clarity and readability of the manuscript can be improved in terms of:

a)        English language

b)       Typographical mistakes (some spaces missing before references)

c)        Consistency

d)       For the first time it is used, an acronym should be explained. After, it is enough to use the acronym

Specific comments:

Line 24: please consider replacing with “case study, using shotgun metagenomics”

Line 26: please consider replacing with “pipelines for recovering ancient microbial DNA information”

Line 28: please consider replacing with “optimized for aDNA research”

Lines 28-29: please consider replacing with “shorter reads (<100 bp) than a commercial kit”

Line 31: please consider replacing with “our data sets and downstream analyses” (remove the comma)

Lines 36 - 38: please consider improving for clarity. For instance “Paleomicrobiological studies of ancient microbial molecules (e. g., DNA, proteins, or lipids) provide insights into how these species, populations, and ecosystems evolve [1–3].”

Lines 38 - 41: please consider improving for clarity. For instanceNowadays, paleomicrobiologists rely on recovering aDNA – a highly degraded, fragmented, and chemically modified DNA - extracted from historical, archaeological, and palaeoecological remains [4].”

Line 55: please consider replacing with “(14.6-12.7 kyr BP). This study suggested”

Line 57: please consider replacing with “in the eastern Australian”

Lines 60 - 61: please consider improving for clarity. For instance “Club Lake. The findings suggested that this region's general warming trend over the late Holocene was concurrent with fire activity and ecosystem shifts.

Line 61: please consider replacing with “The previously presented studies show how microbial aDNA can provide”

Line 75: please consider replacing with “resources of laboratory and bioinformatic”

Line 85: please consider replacing with “analyze environmental microbial aDNA.”

Lines 86 – 90: Confusing text, please consider rephrasing

Line 90: please consider replacing with “Overall, the discussion of the methods presented”

Lines 92 – 93: please consider replacing with “microbial aDNA from environmental samples”

Line 96: please consider replacing with “Terrestrial soil samples”

Line 99: please consider replacing with “availability of crucial information from present and ancient microbial communities”

Line 98 – 101: please consider replacing this sentence, do not belong to the Material and Methods section

Line 124: please consider replacing with “5% sodium hypochlorite and ethanol”

Lines 126 – 129: please consider replacing with “A hole was excavated at each point to collect the samples. The outer layer of soil was removed (approximately 1 cm) using a sterile spatula. The target soil was collected in a 50 mL sterile tube every 20 cm in a vertical transect from the bottom to the top of the excavation hole (Figure 1).”

Line 135: please consider replacing with “1mm yr-1” (superscript)

Line 140: please consider replacing with “six months before DNA extraction”

Line 143: please consider replacing with “Samples containing internal soil were homogenized by mixing.”

Line 144: please consider replacing with “homogenized soil was taken for DNA extraction”

Line 149: please consider replacing with “impact of cross-contamination (EBC2)”

Line 153: please consider replacing with “using an in-house

Line 157: please consider replacing with “and a silica-based method”

Lines 158 – 167: to simplify the text please consider making changes in the text. If the protocol was used exactly performed as described in the bibliographic reference 17, there is no need to describe it. If changes were made, please describe them.

Line 174: please consider replacing with “and mixed under”

Line 175: please consider replacing with “and the pellet was washed”

Line 179: please consider replacing with “incubated for 10 min”

Lines 169 – 180: to simplify the text please consider making changes in the text. If the protocol was used exactly performed as described in the bibliographic reference 34, there is no need to describe it. If changes were made, please describe them.

Lines 181 – 194: to simplify the text please consider making changes in the text. If the protocol was used exactly performed as described in the bibliographic reference 35, there is no need to describe it. If changes were made, please describe them.

Line 184: please consider replacing with “both phases were separated”

Line 185: please consider replacing with “and the process was repeated.”

Lines 188 – 189: please consider replacing with “concentration of 0.3 M.

Lines 189 - 190: please consider replacing with “Then, two volumes of ice-cold ethanol and 0.01 M MgCl2 were added.”

Lines 190 - 191: please consider replacing with “The solution was incubated at -20°C for 30 min and centrifuged at 14,000 rpm, at room temperature, for 15 min.”

Lines 192 - 193: please consider replacing with “The solution was centrifuged at 14,000 rpm, at 4°C, for 2 min, being the supernatant was removed.”

Line 209: please consider replacing with “, and the supernatant was removed”

Line 210: please consider replacing with “min, and resuspended in”

Line 211: please consider replacing with “incubated for 10 min.”

Lines 167/180/195/198/212: please consider removing the sentences and include a single sentence at the end of the description of the extraction methods concerning sample storage. For instance “All extracted DNA samples were aliquoted and stored at -20 ºC until further use.”

Lines 216 – 247: to simplify the text please consider making changes in the text. If the protocol was used exactly performed as described in the bibliographic reference 36, there is no need to describe it. If changes were made, please describe them.

Lines 246 – 247:please consider removing this sentence.

Lines 249 – 289: please consider simplifying the text

Line 250: please consider replacing with “two different pre-processing tools”

Line 257: please consider replacing with “extracted with phenol-chloroform

Line 260: please consider replacing with “and merged using AdapterRemoval v.2.2.1”

Line 262: please consider replacing with “merged using AdapterRemoval v.2.2.1” (remove the comma)

Line 263: please consider replacing with “followed by reads deduplication”

Line 266: please consider replacing with “and merged using AdapterRemoval v.2.2.1.”

Line 268: please consider replacing with “followed by reads deduplication”

Lines 271 – 272: please consider replacing with “A low-complexity threshold of 55% was applied using Komplexity v.0.3.6, followed by reads deduplication”

Line 291: please consider replacing with “on all pre-processed data sets”

Line 293: please consider replacing with “and the total number of sequences”

Line 294: please consider replacing with “both pre-processing software”

Line 296: please consider replacing with “each case obtained from MultiQC”

Lines 299 – 301: please consider replacing with “We examined the impact of the five DNA extractions methods, and eight bioinformatic pipelines for the taxonomic classification of DNA reads recovered from the soil samples.”

Lines 301 - 303: please consider replacing with “Since we used an alignment-based taxonomy classification, we also explored the influence of database choice on the reconstruction of soil microbiota, as it has been shown to bias the results significantly [19,42].

Line 306: please consider replacing with “The taxonomic composition”

Line 308: please consider replacing with “ (default settings and semi-global alignment)”

Lines 313 - 315: please consider replacing with “RMA files were imported into MEGAN6 Community Edition (v.6.19.2 [45]) using the compare function as absolute read counts and ignoring unassigned reads to visualize taxonomic classifications.”

Line 321: please consider replacing with “species were removed from sample data sets”

Line 330: please consider replacing with “and statistical insignificance of p-values > 0.05”

Line 330: please consider replacing with “Beta diversity PCoA was plotted”

Lines 339 - 340: please consider replacing with “we tested four different databases

Line 341: please consider replacing with “chromosome, and scaffold-level” and “database June 2018, containing”

Line 349: please consider replacing with “ (default settings and semi-global alignment)”

Line 355: please consider replacing with “(read counts and ignoring unassigned reads”

Line 362: please consider replacing with “species were removed from sample data”

Line 368: please consider replacing with “with the SILVA database were removed, as they contained fewer classified sequence reads than

Line 370: please consider replacing with “to retain more data sets”

Line 375: please consider replacing with “[46] and statistical insignificance”

Line 376: please consider replacing with “Beta diversity PCoA was plotted”

Line 378: please consider replacing with “were plotted at the genus level”

Line 384: please consider replacing with “HOPS software, we used collapsed”

Line 386: please consider replacing with “(RefSeq, NT, and GTDB)”

Line 388: please consider replacing with “at the species-level”

Lines 389 - 392: please consider replacing with “To simplify the discussion in this article, we selected the results using GTDB as the reference database because we obtained a higher number of mapped reads than Refseq and NT databases (Table S5A).

Line 394: please consider replacing with “and were not considered”

Line 398: please consider replacing with “DNA fragments of varying sizes”

Line 407: please consider replacing with “Contaminant DNA is a primary concern”

Lines 409-410: please consider replacing with “Samples were processed by taking all the precautions listed below to minimize contamination”

Lines 413 – 417: please consider replacing with “Llamas et al. [31] proposed guidelines for sample handling in aDNA studies and recommended several precautions: 1) the use of disposable gloves and changing them between samples; 2) avoiding water to wash samples; and 3) the storage of samples in the cold (-20°C to 4°C) and dry places immediately after the collection to avoid freeze/thaw cycles.”

Lines 428 – 429: please consider replacing with “ (reviewed in [31,54–59])

Line 432: please consider replacing with “(>1.45 J cm-2))” (superscript)

Line 438: please consider replacing with “Modern samples should not be processed”

Lines 442 – 443: please consider replacing with “ To remove or minimize pre-laboratory surface contamination, sample decontamination prior to subsampling and DNA extraction is required.”

Line 445: please consider replacing with “and sodium hypochlorite) routinely applied to bones”

Line 450: please consider replacing with “sediment cores are removed”

Line 459: please consider replacing with “mixing the original”

Line 467: please consider replacing with “should also be monitored by introducing extraction”

Lines 474 – 475: please consider replacing with “using the strict cleaning procedures described above”

Lines 481 - 482: please consider replacing with “especially with HTS techniques [72]”

Line 486: please consider replacing with “environmental DNA while reducing”

Lines 490 - 493: please consider replacing with “Extraction protocols for aDNA have been primarily standardized using human and animal archaeological remains. Only a few protocols have been optimized for the recovery of microbial aDNA, especially from environmental samples.”

Lines 506 - 507: please consider replacing with “than the commercial kit. However, all the protocols showed a consistent taxonomic profile”

Lines 508 - 511: please consider replacing with “After DNA extraction, the next step is the preparation of sequencing libraries. Currently, two DNA-based approaches are typically used to characterize the ancient environmental microbiomes: amplicon sequencing and shotgun metagenomics.”

Line 523: please consider replacing with “a wide range of calculus samples”

Lines 524 - 525: please consider replacing with “A bioinformatic analysis pipeline processes sequencing data to reconstruct the soil microbiome.”

Line 526: please consider replacing with “Pre-processing raw HTS sequence”

Line 527: please consider replacing with “downstream analysis and comprises a series of steps” (remove the comma)

Lines 528 - 530: please consider replacing with “As HTS of short DNA fragments may result in the incorporation of adapter sequences into the data set, contaminating the dataset and negatively impacting the analysis [37], adapter contamination must first be removed.”

Lines 530 - 531: please consider replacing with “Second, if applicable, collapsing (merging) paired-end reads can be applied.”

Line 532: please consider replacing with “detecting overlapping reads”

Line 537: please consider replacing with “are used to eliminate other HTS biases”

Lines 539 - 540: please consider replacing with “is used assuming that PCR amplification”

Lines 549 - 550: please consider replacing with “After pre-processing, the next step is identifying the microbial taxa in ancient samples.”

Line 556: please consider replacing with “observed that nucleotide-to-nucleotide alignments”

Line 563: please consider replacing with “of human-associated microbes”

Lines 564 - 565: please consider replacing with “microbes [19,90]. However, this effect is likely much higher in environmental studies with fewer reference sequences.”

Line 567: what does (e.g., 17, 19, 33, 93, 94) refer to?

Line 568: please consider replacing with “We integrate these concepts”

Lines 578 - 579: please consider replacing with “we retained 5,685 - 30.6 M collapsed reads”

Line 587: please consider replacing with “read length were observed between”

Line 590: please consider replacing with “showed significantly higher retention”

Line 593: please consider replacing with “Table S1C) and did not show” (remove the comma)

Lines 594 - 595: please consider replacing with “In general, the most significant effects on sequence pre-processing

Lines 613 - 615: please consider replacing with “during the purification of small-sized nucleic acid molecules from human urine”

Lines 615 - 616: please consider replacing with “and phenol-chloroform recovered longer fragments”

Line 621: please consider replacing with “should be cautioned.”

Lines 629 - 630: please consider replacing with “PowerSoil kit, phenol-chloroform method”

Line 631: please consider replacing with “for the phenol-chloroform method”

Lines 635 - 636: please consider replacing with “and phenol-chloroform, indicating a lower contribution of cross-contamination

Lines 637 - 638: please consider replacing with “presented higher reads in EBC2s than EBC1s”

Line 641: please consider replacing with “suggest that minor contaminant species’ DNA”

Line 647: please consider replacing with “ability to characterize microbial communities accurately [47,70].”

Lines 647 - 649: please consider replacing with “For this reason, contaminant species must be filtered from data sets before analyzing a sample's taxonomic composition and diversity.

Line 657: please consider replacing with “A high proportion of unassigned”

Line 658: please consider replacing with “likely due to unknown species”

Line 660: please consider replacing with “at the species level”

Line 664: please consider replacing with “and phenol-chloroform methods”

Line 666: please consider replacing with “the sequences were probably recovered”

Line 670: please consider replacing with “sequences than collapsed reads”

Line 671: please consider replacing with “suggesting higher contamination levels

Line 677: please consider replacing with “than in post-decontaminated samples”

Line 682: please consider replacing with “whether the removal of contaminant species”

Lines 684 - 686: please consider replacing with “As expected, every decontaminated sample resulted in lower observed features than contaminated samples (Table 685 S3A).”

Lines 688 – 693: Confusing text, please consider rephrasing.

Lines 699 – 670: please consider replacing with “ancient studies due to the low concentration” (remove the comma)

Line 706: please consider replacing with “each sample was considerably different”

Line 710: please consider replacing with “were more significant in domains”

Line 712: please consider replacing with “using phenol-chloroform extraction”

Line 715: please consider replacing with “using the phenol-chloroform method”

Lines 716-717: please consider replacing with “Overall, the extraction method impacted

Line 727: please consider replacing with “suggests that a more significant number”

Lines 730 - 733: please consider replacing with “For example, alpha diversity (observed features and Shannon’s diversity indices) slightly decreased in samples extracted with methods with poor DNA yields compared to samples with a higher number of collapsed reads (Table S3A).

Lines 733 - 734: please consider replacing with “in the features index between extraction protocols”

Lines 739 - 740: please consider replacing with “but the extraction method also impacts the composition”

Lines 742 - 745: please consider replacing with “samples [15], we show that DNA signals recovered from environmental samples are potentially more susceptible to contamination introduced by modern microbes living in these environments and laboratory protocols”

Lines 747 - 749: please consider replacing with “Overall, we recommend using the SiO2 + PB buffer and SiO2 + QG buffer methods to recover short microbial DNA sequences from ancient soil samples, according to their performance in this case study and previous aDNA studies (e.g., [6]).”

Line 765: please consider replacing with “Although ribosomal RNA (rRNA) genes” (remove the comma)

Line 768: please consider replacing with “low memory consumption

Line 769: please consider replacing with “AdapterRemoval2 vs. Fastp”

Line 788: please consider replacing with “the reconstructed microbial communities' clustering was primarily driven by”

Line 815: please consider replacing with “diversity analysis comparison, we removed SILVA” (add comma)

Line 816: please consider replacing with “databases' data. Further, comparing taxonomic profiles”

Line 820: please consider replacing with “was found using the SILVA SSU 132 database”

Line 822: please consider replacing with “50% of the classified sequences”

Line 824: please consider replacing with “represented in black in Figure S4”

Line 826: please consider replacing with “extracted with phenol-chloroform)”

Lines 835 - 836: please consider replacing with “associated with databases for reconstructing ancient metagenomes”

Lines 841 - 842: please consider replacing with “environmental samples than human-associated samples”

Line 845: please consider replacing with “HTS, several publications lack”

Line 848: please consider replacing with “damage patterns consistent with” and “suggested nucleotide misincorporation”

Line 852: please consider replacing with “correlated with the sample's age

Lines 854 - 856: please consider replacing with “These tools have verified that microbial DNA degrades at appreciable rates, although they may be different across species with different G/C ratios or slower than rates observed in humans [42,106].”

Lines 857 - 858: please consider replacing with “can also be used to validate ancient sequences using phylogenetic analysis, as ancient genetic variation should fall outside that seen today [31,54]”

Lines 859 - 860: please consider replacing with “have been developed to identify a single species of interest (e.g., humans) and not a complex dataset with a high risk of contamination”

Lines 867: please consider replacing with “bioinformatics protocols by testing” (remove the comma)

Lines 867 - 868: please consider replacing with “using two statistical models”

Lines 874 - 876: please consider replacing with “However, reference genome biases can heavily impact alignment-dependent authentication tools, particularly when working with rare and poorly studied taxa, such as those found in extreme environments.”

Lines 878 - 880: please consider replacing with “To address these issues, we also included a second authentication model (Changepoint) based on an alignment-free algorithm that does not require a reference genome [51].”

Lines 890: please consider replacing with “have a continuous decline of C to T”

Lines 892 - 894: please consider replacing with “Mapping to an incorrect species generally results in increased mismatches across the whole read, which is evident in the edit distance distribution [50].”

Lines 898 - 899: please consider replacing with “we compared the damage in Acidobacteria bacterium (the only species represented across all data sets”

Lines 901: please consider replacing with “SiO2+PB buffer than any other”

Lines 904 - 905: please consider replacing with “distance plots with a continuous decline”

Line 910: please consider replacing with “consistent with previous observations”

Lines 911 - 912: please consider replacing with “The results obtained with the software HOPS and Changepoint were consistent.”

Lines 914 – 916: please consider removing this sentence-

Line 940: please consider replacing with “Ancient DNA has helped to describe”

Lines 942 - 943: please consider replacing with “have proliferated mainly in the last decade”

Line 948: please consider replacing with “aDNA signal compared to commercial kits and phenol-chloroform method”

Line 955: please consider replacing with “examine the cellular degradation”

Author Response

Reply to reviewer 2: We thank the reviewer for their careful, insightful review of our manuscript, as well as their valuable feedback. We incorporated all of these changes provided by the reviewer, and where a discrepancy existed, we provide our response below. We tracked changes within the revised manuscript.

Specifically, we want to thank the reviewer for suggesting further improvements in clarity and readability of the manuscript. Grammatical errors have been checked and corrected by native English speakers, and we carefully addressed the typographical mistakes. We also double-checked the consistency of the manuscript, and modified the manuscript accordingly.

For example, we now use “microbial aDNA” throughout the manuscript, instead of “microbial ancient DNA” (e.g., Line 23) or ancient microbial DNA (e.g., Line 26).

Lastly, we now define the acronyms when they appear the first time and use the acronyms directly in the main text, including:

Line 17: High Throughput DNA Sequencing (HTS)

Line 18: ancient DNA (aDNA)

Line 182: extraction blank controls (EBCs) 

Line 227: Phenol chloroform (PHCH)

Line 242: Powerlyzer kit (PL)

Line 245: SiO2 + QG buffer (QG)

Line 400: MEGAN Alignment Tool (MALTn)

Line 452: lowest common ancestor (LCA)

Line 495: Heuristic Operation for Pathogen Screening (HOPS)

Line 998: short-sequence DNA repeats (SSR)

The suggestions below were not incorporated into the revised version:

Lines 167/180/195/198/212: please consider removing the sentences and include a single sentence at the end of the description of the extraction methods concerning sample storage. For instance “All extracted DNA samples were aliquoted and stored at -20 ºC until further use.”

Reply: We thank the reviewer for this suggestion. However, we believe the descriptions of the protocols are more precise for the reader when we add this sentence at the end of each section.

Line 710: please consider replacing with “were more significant in domains”

Reply: Thank you for the suggestion. We did not use ‘more significant’ because we could not apply statistical analysis here. Samples with low number of reads and poor DNA yield were eliminated during the rarefaction step, thus significant differences of taxonomic profiles between these extraction methods were not included.